# PDP: Parameter-free Differentiable Pruning is All You Need

**Minsik Cho**      **Saurabh Adya**      **Devang Naik**

Apple Inc.

{`minsik, sadya, naik.d`}@apple.com

## Abstract

DNN pruning is a popular way to reduce the size of a model, improve the inference latency, and minimize the power consumption on DNN accelerators. However, existing approaches might be too complex, expensive or ineffective to apply to a variety of vision/language tasks, DNN architectures and to honor structured pruning constraints. In this paper, we propose an efficient yet effective train-time pruning scheme, Parameter-free Differentiable Pruning (PDP), which offers state-of-the-art qualities in model size, accuracy, and training cost. PDP uses a dynamic function of weights during training to generate soft pruning masks for the weights in a parameter-free manner for a given pruning target. While differentiable, the simplicity and efficiency of PDP make it universal enough to deliver state-of-the-art random/structured/channel pruning results on various vision and natural language tasks. For example, for MobileNet-v1, PDP can achieve 68.2% top-1 ImageNet1k accuracy at 86.6% sparsity, which is 1.7% higher accuracy than those from the state-of-the-art algorithms. Also, PDP yields over 83.1% accuracy on Multi-Genre Natural Language Inference with 90% sparsity for BERT, while the next best from the existing techniques shows 81.5% accuracy. In addition, PDP can be applied to structured pruning, such as N:M pruning and channel pruning. For 1:4 structured pruning of ResNet18, PDP improved the top-1 ImageNet1k accuracy by over 3.6% over the state-of-the-art. For channel pruning of ResNet50, PDP reduced the top-1 ImageNet1k accuracy by 0.6% from the state-of-the-art.

## 1 Introduction

Deep neural networks (DNN) have shown human performance on complex cognitive tasks [46], but their deployment onto mobile/edge devices for enhanced user experience (i.e., reduced latency and improved privacy) is still challenging. Most such on-device DNN systems are battery-powered and heavily resource-constrained, thus requiring high power/compute/storage efficiency [22, 48, 50, 53].

Such efficiency can be accomplished by mixing and matching various techniques, such as designing efficient DNN architectures like MobileNet/MobileViT/ MobileOne [36, 43, 48], distilling a complex DNN into a smaller architecture [41], quantizing/compressing the weights of DNNs [8, 18, 23, 31, 39, 58], and pruning near-zero weights [28, 34, 40, 44, 52, 56, 57, 61]. Also, pruning is known to be highly complementary to quantization/compression [51] when optimizing a DNN model. Training a larger model and then compressing it by pruning has been shown to be more effective in terms of model accuracy than training a smaller model from the beginning [32]. However, pruning comes at the cost of degraded model accuracy, and the trade-off is not straightforward [28].

Hence, a desirable pruning algorithm should achieve high accuracy and accelerate inference for various types of networks without significant training overheads in costs and complexity. In this

37th Conference on Neural Information Processing Systems (NeurIPS 2023).

| | STR$^a$ | FTWT$^b$ | CS$^c$ | GraNet$^d$ | OptG$^e$ | ACDC$^f$ | MVP$^g$ | POFA$^h$ | PDP |
|---|---|---|---|---|---|---|---|---|---|
| Conv-Net Accuracy | ✓ | ✓✓ | ✓✓ | ✓ | ✓✓ | ✓✓ | ? | ? | ✓✓✓ |
| Transformer Accuracy | ✓ | ? | ? | ? | ✓ | ? | ✓✓ | ✓✓ | ✓✓✓ |
| Inference speed | ✓✓✓ | ✓✓✓ | ✓✓✓ | ✓ | ✓✓✓ | ✓ | ? | ? | ✓✓✓ |
| Training speed | ✓✓✓ | ? | ✓ | ✓✓ | ✓✓ | ✓ | ? | ? | ✓✓✓ |
| Training stability | ✓ | ? | ✓ | ✓✓ | ✓✓✓ | ✓✓✓ | ? | ? | ✓✓✓ |
| Training flow | simple | complex | complex | complex | complex | complex | complex | complex | simple |
| Extra parameters | a few | many | many | none | many | none | many | none | none |

$^a$ Soft threshold weight re-parameterization for learnable sparsity [28].
$^b$ Fire Together Wire Together: A Dynamic Pruning Approach with Self-Supervised Mask Prediction [11]
$^c$ Winning the Lottery with Continuous Sparsification [45]
$^d$ Sparse training via boosting pruning plasticity with neuro–regeneration [34].
$^e$ Optimizing gradient-driven criteria in network sparsity: Gradient is all you need [57].
$^f$ AC/DC: alternating compressed/decompressed training of deep neural networks [40].
$^g$ Movement Pruning: Adaptive Sparsity by Fine-Tuning [44].
$^h$ Prune Once for All: Sparse Pre-Trained Language Models [56].

**Table 1:** Comparison of the state-of-the-art pruning schemes and PDP: PDP can explore a good trade-off bewteen accuracy and inference speed without introducing new learnable parameters and with a simple/fast training flow.

work, we propose a simple yet effective pruning technique, Parameter-free Differentiable Pruning or PDP, which uses a dynamic function of weights to generate soft pruning masks for the weights themselves. PDP requires neither additional learning parameters [57] nor complicated training flows [40], yet offers precise control on the target sparsity level [28], while pushing the state-of-the-arts in random/structured/channel pruning. The soft masks from PDP make pruning differentiable and let training loss decide whether/how each weight will be pruned [42]. Table 1 compares PDP with the latest state-of-the-art pruning schemes, and our major contributions include:

- PDP outperforms the state-of-the-art schemes on a variety of models and tasks. Being differentiable and parameter-free, PDP offers efficient and effective pruning without complex techniques.

- PDP offers a universal and holistic approach for efficient random/structured/channel pruning, while delivering a high-quality model optimization for a given pruning target.

- With a dynamic function of weights, PDP generates a soft pruning mask which does not need training, and thus does require neither gradient synchronization nor SGD-update.

## 2   Related Works

Pruning in DNN incurs a complex trade-off between model accuracy and inference speed in terms of MAC (mult-add operations) [28]. A weight can contribute differently to the model accuracy, depending on the number of times it is used for prediction (i.e., a weight in convolution filter for a large input) and the criticality of the layer it belongs to (i.e., a weight in a bottleneck layer). Therefore, even if two models are pruned to the same level, the accuracy and inference speed of each can be vastly different, which makes exploring the best trade-off challenging yet crucial in DNN pruning. A body of work in random and structured pruning has been proposed to optimize such a trade-off. Please see Fig. 8 and Section B in Appendix for additional reviews and details.

Differentiable techniques gained popularity due to the advances in network architecture search methods as well as network compression techniques [6, 8, 33], and is a powerful technique for random/structured pruning as it drives pruning based on task loss. However, differentiable pruning does not necessarily deliver the best quality pruning (i.e., many state-of-the-art results are from non-differentiable methods), because the computational overhead and training complexity can limit its benefit. Existing differentiable pruning methods can be classified into a few approaches.

**Learning pruning budget allocation** is to determine target sparsity for each layer in a differentiable way, rather than computing pruning masks. Optimizing and training a per-layer pruning threshold with ReLU based on dynamic re-parameterization was proposed to allocate the sparsity across all layers in STR [28], which shows good model accuracy and low inference overhead. DSA [38] finds the layer-wise pruning ratios in a differentiable fashion by computing a channel-wise keep probability drawn from the Bernoulli distribution and keeping the expected sparsity ratio satisfied.

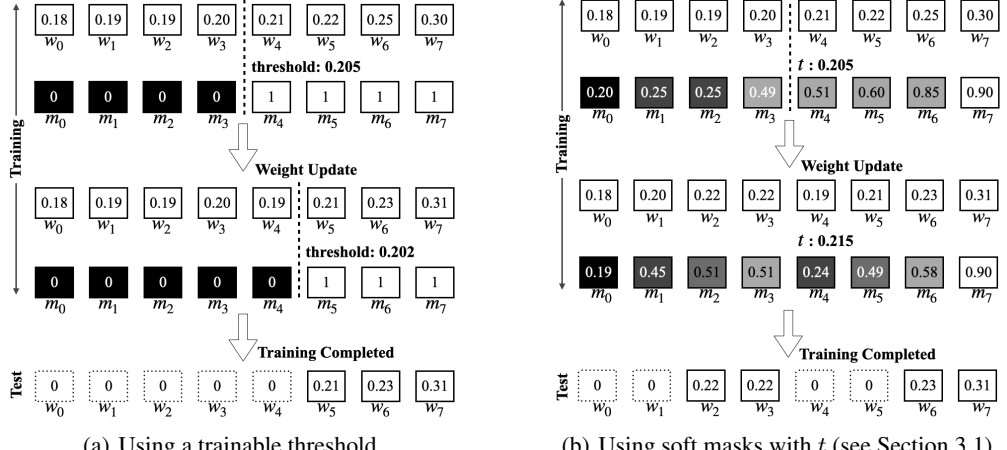

(a) Using a trainable threshold

(b) Using soft masks with $t$ (see Section 3.1)

**Figure 1:** Unlike learning a threshold parameter for differentiable pruning [28, 38] in (a), PDP generates soft masks without extra trainable parameters to accomplish differentiable pruning as in (b). The moment a weight is pruned in (a), the weight will not get any update as the mask zeros out the gradient, depriving the opportunity for better model training. However, the scheme in (a) ensures the training and test behaviors are identical, yet makes the sparsity control difficult. PDP generates such soft masks without extra parameter/training overheads unlike existing schemes [34, 40, 44, 45, 57] (see Section 3.1). While PDP reveals different behaviors during training/test times because the test needs hard masks, our soft mask allows a weight to recover from being pruned over time, which yields higher model accuracies efficiently.

**Learning pruning mask** as an extra parameter is a popular way for differentiable pruning. In CS [45], a new trainable mask parameter is learned to continuously remove sub-networks based on the lottery hypothesis [13]. In detail, the new mask parameter is first multiplied by a scheduled temperature parameter, and then L0-regularized inside the Sigmoid function to continuously increase the sparsity. OptG [57] proposed learning a mask parameter based on the loss changes due to pruning a particular weight, which is proportional to the gradient and the magnitude of the corresponding masked weight when approximated with STE [5]. AutoPrune [54] also introduces extra learnable parameters (controlled by a regularizer) to generate a differentiable/approximated pruning mask using STE [5]. In [42], new learnable parameters, $\alpha$ are introduced and $L_1$-regularized to enforce sparsity. Then, $\alpha$ is compared with a hyper-param $t$ to derive the masks for the model parameters (i.e., prune if $\alpha < t$). As such comparison is not differentiable, comparison is approximated into a foothill function [4]. Learning channel pruning mask is proposed in SCP [25] based on the operation-specific observation that a feature map with a large and negative batch mean will get zeroed by ReLU .

**Generating pruning masks** is proposed in FTWT [11] where new layers are attached to the exiting main architecture. The attached new layers are fed with activations from the main network, then trained to generate masks for the weights in the main network. Thus, although it does not learn masks directly, it still requires new trainable parameters to learn how to generate masks, not to mention the changes to the model network. Using knowledge distillation to generate pruning masks is explored in NISP [55] and DCP [62], where a fully trained teacher network guides the mask generation in a way that the distortion to the layer responses can be minimized. A differentiable Markov process is studied to generate a channel pruning mask probabilistically, where a state represents a channel and the transitions from states account for the probability of keeping one channel unpruned [16].

## 3 PDP: Parameter-free Differentiable Pruning

Complex pruning schemes do not always yield the best quality results, and their complexity and cost can make them impractical and difficult to use. The proposed Parameter-free Differentiable Pruning (PDP) is a highly effective and efficient scheme that generates soft pruning masks using a dynamic function of weights in a parameter-free and differentiable fashion. Since PDP is differentiable, the task loss can directly guide the pruning decision, offering an effective pruning solution. Simultaneously, being parameter-free, PDP can be fast and less intrusive to the existing training flow. Overall, PDP finds a weight distribution that is best for task loss and pruning. Instead of having extra parameters, PDP indirectly influences the weight distribution for high-quality pruning. For example, if a weight

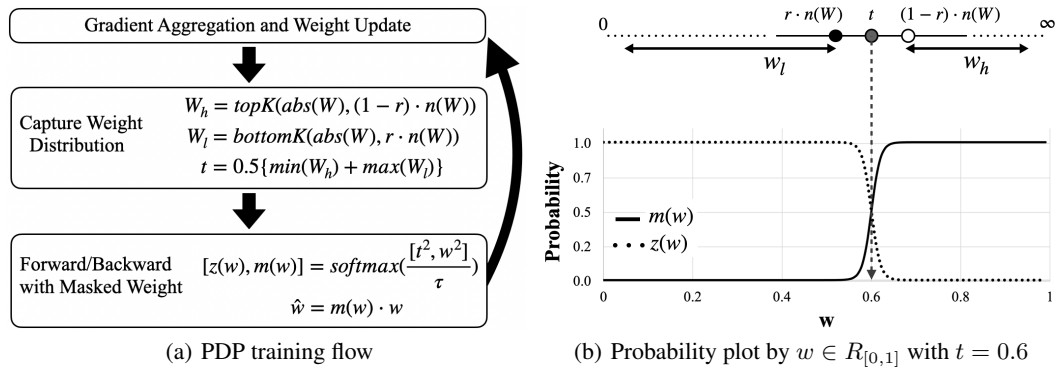

(a) PDP training flow        (b) Probability plot by $w \in R_{[0,1]}$ with $t = 0.6$

**Figure 2:** Computing $z(w), m(w)$ for the chances for $Z$ and $M$ with $t$ for the equal chance to be in $Z$ and $M$.

$w$ is destined to be pruned for some reason, instead of having a new parameter to denote **"to-prune"**, PDP lets SGD gradually make $w$ itself smaller relatively against other parameters in the same entity, increasing its chance to be pruned over time. We will first present PDP in Section 3.1, provide in-depth analysis in Section 3.2, discuss the benefits of PDP over existing differentiable pruning approaches in Section 3.3, and then show the extension to structured and channel pruning in Section 3.4.

### 3.1 PDP Algorithm

To address the drawbacks of existing differentiable pruning algorithms, we propose PDP. A soft mask should ideally represent the chance of a weight $w$ being in one of two symbolic states, "to-prune" (noted as $Z$) or "not-to-prune" (noted as $M$), without requiring extra parameters or expensive book-keeping. While the chance of $w$ being in either state is not straightforward to compute, PDP generates a soft mask based on the fact that there exists an equal chance point for both states. Let us consider differentiable functions, $z, m : \mathbb{R}[0, \infty] \mapsto \mathbb{R}[0, 1]$, to compute the chances of being in $Z$ and $M$, respectively, which must satisfy the following conditions as a soft mask for magnitude-based pruning:

- $z(|a|) < z(|b|)$ for $|a| > |b|$: a weight with a smaller magnitude has a higher chance to be in $Z$.

- $m(|a|) > m(|b|)$ for $|a| > |b|$: a weight with a larger magnitude has a higher chance to be in $M$.

- $z(w) + m(w) = 1$ for any $w$: the total probability is 1.

$$z(w) = \begin{cases} 1 & \text{if } w = 0 \\ 0 & \text{if } |w| = \infty \\ \frac{1}{2} & \text{if } |w| = t \end{cases}$$

Then, by the monotonicity and continuity, there exists $t \in \mathbb{R}_{\geq 0}$ such that $z(t) = m(t) = 0.5$ (the equal chance for $Z$ and $M$), which leads to the following boundary conditions on the left. Any function that satisfies these conditions can be used to compute $m(w)$ as a soft mask of $w$ for train-time pruning. Let denote that $topK(X, k)/bottomK(X, k)$ is selecting the largest/smallest $k$ elements from a matrix $X$, $abs(X)$ is an element-wise absolute operation, and $n(X)$ returns the element count. In PDP, we uniquely identify $t$ for a given prune ratio $r \in [0, 1)$ for a layer with a weight matrix $W$ as in Fig. 2 (a) based on the pruning context.

- The sparsity $r$ for each $W$ can be easily obtained by sorting the weights from the network per magnitude after a few epochs w.r.t the global target sparsity as a one-time task, by a pruning budget allocation algorithm [14, 28], or set manually by a user.

- Right after the SGD weight update, $t$ is computed for the weights $W$ in each layer or entity. The role of $t$ is to abstract the current weight distribution of each layer/entity for pruning.

- During forward-pass, a soft mask, $m(w)$ for the weight $w$ is computed and then the masked weight $\hat{w}$ is applied. $\tau$ is the temperature parameter (see Section E in Appendix for details).

- Computing $t$ and generating $\hat{w}$ repeat iteratively to adapt to the updated weight distribution.

Figure 2 (b) shows how the value of $t$ is obtained in PDP and a soft mask is computed. Specifically, $t$ is set to the value that is halfway between the largest pruned weight and the smallest unpruned weight when a hard mask is applied for a given sparsity ratio $r$. This ensures that each weight has an equal probability of being pruned or kept. As a result, PDP satisfies all the constraints for $z$ and $m$. More details on PDP training flow are in Section D in Appendix.

PDP uses a dynamic function of $t$ to generate soft pruning masks of $W$ without the need for any extra trainable parameters. Instead, PDP lets the weights of the network adjust themselves such that the information that would otherwise be learned by the extra trainable parameters is instead fused into the weights themselves and their distribution. This is possible because each weight $w$ is not only a coefficient in a layer, but also an indicator of the relative chance of that weight being pruned against the other weights in $W$. This relative chance is determined by the value of $t$, as shown in Figure 2 (b). Overall, PDP looks simple but is shown to be quite effective, which opens to broad applicability.

### 3.2 PDP Gradient Analysis

To better understand how PDP helps pruning decision, we derive the gradient of the masked weight $\hat{w}$. As in Fig. 2 (a), $\hat{w}$ is the following:

$$\hat{w} = m(w) \cdot w = \frac{e^{\frac{w^2}{\tau}}}{e^{\frac{w^2}{\tau}} + e^{\frac{t^2}{\tau}}} \cdot w \tag{1}$$

Then, the gradient of $w, \Delta w$ can be simplified as the next:

$$\Delta w = m(w)\Delta\hat{w} + 2\frac{w^2}{\tau}m(w)\{1 - m(w)\}\Delta\hat{w} \tag{2}$$

where we can make the following observations:

- The 1st term is a typical gradient in mask-based pruning, and the 2nd term is an additional gradient with a positive factor (i.e, $\frac{w^2}{\tau}m(w)\{1 - m(w)\} \geq 0$) from PDP.
- If $m(w) \approx 0$ or hard-prune, $\Delta w \approx 0$, which is true for any pruning algorithm.
- If $m(w) \approx 1$ or hard-not-to-prune, $\Delta w \approx \Delta\hat{w}$, which is true for any pruning algorithm.
- When $m(w) \approx 0.5$ (i.e., pruning decision is unclear), $m_w(1 - m_w)$ is maximized and accordingly $\Delta w$ is too, boosting the $w$ movement.
- $\tau$ serves as an inverse scaling factor for the boosted gradient.

Hence, PDP will accelerate the SGD updates for the weights near the pruning boundary (i.e., $t$) toward a loss-decreasing direction, which can encourage the weights to settle with the proper pruning decision at the end. Even if the current gradient is not globally beneficial for the task, many *second* chances will eventually help recover the damages in an accelerated manner. We conjecture these features enable PDP to make a better pruning decision for the weight on the boundary.

### 3.3 PDP vs. Existing Differentiable Pruning Strategies

**Learning Pruning Budget Allocation** focuses on obtaining a pruning threshold in a differentiable way based on marginal loss or $L_1$-regularization [25, 38]. Meanwhile, PDP directly generates a soft pruning mask for each weight also in a differentiable way. Fig. 1 illustrates the differences with an example. Learning a pruning threshold is helpul for global pruning budget allocation, as the threshold gets adjusted per the task loss as in Fig. 1 (a), but has the following drawbacks:

- Sparsity level is hard to control, as the pruning threshold is not explicitly related to the sparsity, as masks and the thresholds are not directly co-optimized. In Fig. 1 (a), after one weight update, the threshold is reduced from 0.205 to 0.202, but the sparsity is increased from 50% to 66% as $w_4$ becomes smaller than the threshold.
- Once a weight gets pruned during training, it does not get updated as the gradient is masked out. The bottom weights $w_{\{0,1,2,3\}}$ get no weight update due to the zero masks, and remain pruned.

On the contrary, PDP allows all the weights to be updated through soft masks as in Fig. 1 (b), providing higher flexibility and recovery from undesirable pruning [17]. For example, consider the weight $w_2$ of a value 0.19 which would have been permanently pruned in (a). PDP discourages a weight from being permanently pruned through the weight update, $w_2$ still receives a scaled-down gradient. If a near-zero weight continues to get negative gradients over time (although scaled down by the soft mask), it can eventually get unpruned at the end of training. Similarly, if a very large weight gets positive gradients many times enough to be near zero, it will get pruned eventually, even

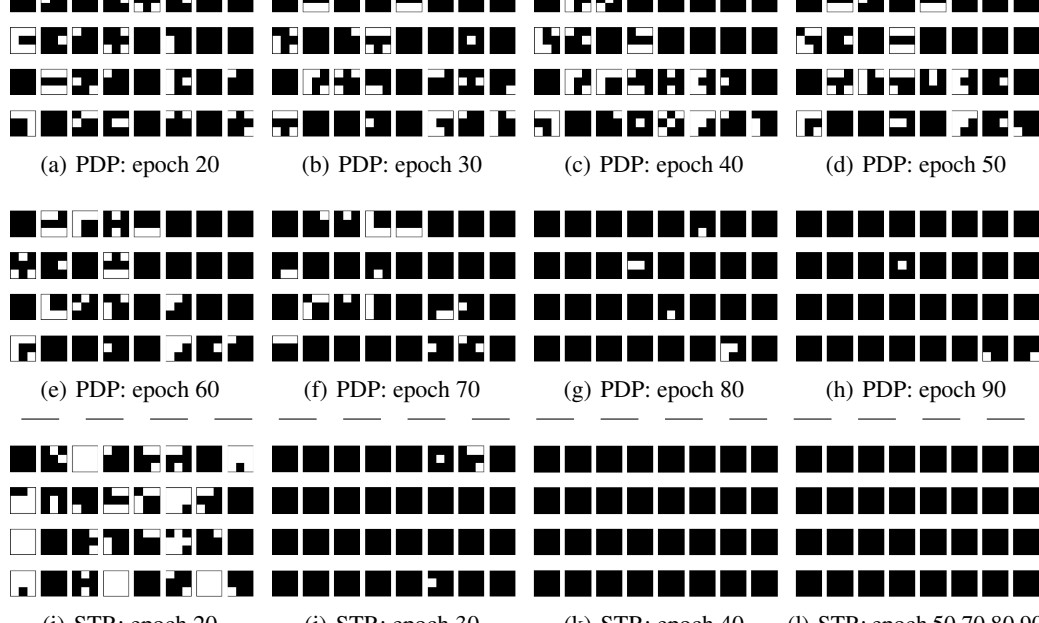

(a) PDP: epoch 20    (b) PDP: epoch 30    (c) PDP: epoch 40    (d) PDP: epoch 50

(e) PDP: epoch 60    (f) PDP: epoch 70    (g) PDP: epoch 80    (h) PDP: epoch 90

(i) STR: epoch 20    (j) STR: epoch 30    (k) STR: epoch 40    (l) STR: epoch 50,70,80,90

**Figure 3:** The effects of PDP and STR [28] (from Table 1) for the first 3x3 Conv2d layer with 32 filters in MobileNet-v1 on ImageNet1k where each small rectangle represents one 3x3 kernel: For PDP, during the entire end-2-end training, we temporarily round the soft mask value to make the pruning decisions. The white cell indicates such pruning decision for the corresponding weight has been flipped at least once during the particular epoch, and the black cell indicates the other case.

if it was not pruned in the early stage. Hence, such gradual pruning decision over time during training allows PDP to make better pruning decisions w.r.t. the task loss.

Such *soft* masking allows pruning decisions to be flipped multiple times during the entire training process, and such effects are compared between PDP and STR [28] (from Table 1) in Fig. 3 where the pruning decisions for the first Conv2d (3x3 kernels with 32 output channels) in MobileNet-v1 for the end-to-end ImageNet1k training are captured. The white cell means that the pruning decision has been flipped (from to-prune to not-to-prune or vice-versa), while the black cell means the decision has never changed, during the given epoch. Then, we can observe from Fig. 3 that while PDP in (a)-(h) continues to flip the decisions until the very late training stage, STR in (i)-(l) finalizes the pruning decisions early (i.e., in the first 30 epochs).

**Learning/Generating Pruning Masks with Extra Parameters** allows the pruning decision to be driven by a task loss through back-propagation rather than the weight value itself (i.e., a hard mask will zero out the gradient of a pruned weight), thus addressing the problems with differentiable pruning budget allocation, but comes with the following issues.

- A pruning mask is (or derives from) a learnable parameter. Hence the number of total trainable parameters increases significantly, making the training process slow and complex [11, 44, 45, 57].

- A hard mask is approximated into a soft mask using a differentiable function, without guaranteeing the key properties of a pruning mask, such as the [0,1] value range or monotonicity [42].

Table 2 compares parameter-free PDP with a differentiable mask pruning scheme, CS [45] from Table 1 (see Table 8 in Appendix for detail). The results show that PDP outperforms CS and is 2.8 faster, without adding extra trainable parameters or complicating the training flow. Table 3 also demonstrates being parameter-free can provide the substantial training efficiency gains for very large language models like GPT. When trained and sparsified on 32 GPUs in mixed-precision (to fit OptG into GPU memory) using the recipe in [1], PDP delivers the best perplexity at 75% sparsity with 42% lower cost than OptG [57].

| Method | Top-1 (%) | Sparsity (%) | Model #param | Extra #param | Runtime min | Inference MAC($\times e6$) |
|--------|-----------|--------------|--------------|--------------|-------------|---------------------------|
| Dense | 91.4 | 0 | 273k | 0 | 22.1 | 41.0 |
| CS | 89.1 | 86.3 | 273k | 267k | 84.7 | 5.80 |
| PDP | 90.4 | 86.3 | 273k | 0 | 30.3 | 5.79 |

**Table 2:** Compared with CS for ResNet20/CIFAR10, PDP delivers 2.7x speed up due to no extra parameters.

| Method | Perplexity | Model #param | Extra #param | GPU cost($) |
|--------|-----------|--------------|--------------|-------------|
| Dense | 22.4 | 163M | 0 | |
| GMP [61] | 37.7 | 163M | 0 | 6997 |
| OptG | 33.7 | 163M | 124M | 11210 |
| PDP | 33.7 | 163M | 0 | 7499 |

**Table 3:** With 75% sparsity, being parameter-free becomes more important for training GPT2 with OpenWebText.

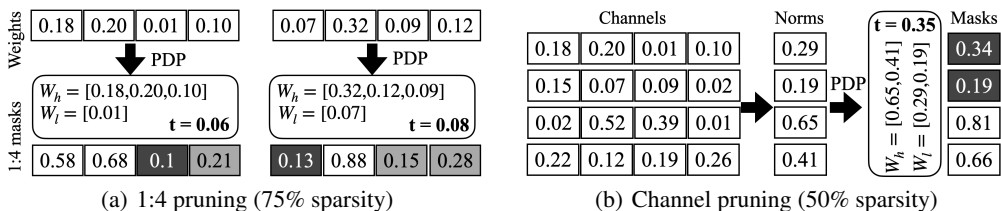

(a) 1:4 pruning (75% sparsity)    (b) Channel pruning (50% sparsity)

**Figure 4:** PDP is simple and universal enough to be applied directly to structured and channel pruning.

### 3.4   PDP for Structured and Channel Pruning

The simplicity and non-intrusive nature of PDP make it readily applicable to differentiable structured and channel pruning. As an example of structured pruning, we consider N:M pruning, where only $N$ weights are kept out of every $M$ consecutive weights. N:M pruning is attracting high research and industrial attention because top-of-the-line GPUs support 2:4 configuration [24]. To apply PDP to N:M pruning, we apply it to every $M$ consecutive weights of the layer, as if the layer were composed of many sub-layers, each with $M$ weights, which is illustrated in Figure 4 (a). Since N:M dictates the target sparsity, we can easily find the threshold $t$ and generate the soft mask, as shown in Figure 2 (a).

Channel pruning is another type of structured pruning that can be easily applied to PDP with minor modifications. To do this, we first compute the $L_2$ norm of each channel in the layer, and then use these norm values (in place of the absolute values of the weights in Figure 2(a)) to compute a soft mask for each channel, which is depicted in Figure 4 (b). Using the soft mask to prune all the corresponding weights in the channel will make the channel pruning process differentiable.

## 4   Experimental Results

We compared our **PDP** with state-of-the-art random, structured, and channel pruning schemes on various computer vision and natural language models. We used two x86 Linux nodes with eight NVIDIA V100 GPUs on each in a cloud environment. All cases were trained from scratch. More experimental results and the hyper-parameters are in Section F and Table 8 in Appendix.

**Random Pruning for Vision Benchmark:** We compared the proposed **PDP** with the latest prior arts, **STR** [28], **GMP** [61], **DNW** [52], **GraNet** [34], **OptG** [57], and **ACDC** [40] on ResNet18, ResNet50, MobileNet-v1, and MobileNet-v2 [20, 22, 43] with the ImageNet1k dataset [9]. Since all of these schemes have been experimented only with ResNet50 and/or MobileNet-v1, we reproduced the pruning results in our controlled environment with the identical data augmentations by running the official implementations from the authors [28, 34, 40, 57] or verified implementations from the prior arts [52, 61] as in Section G in Appendix. Since the primary goal of pruning is to trade-off the model accuracy with the compute reduction as in Section 2, we measured the accuracies and Multiply-Accumulate Operation (MAC) during inference on each experiment with layer fusion (i.e., BatchNorm folding), and mainly focused on the high-sparsification cases. Note that the MAC is purely theoretical and reported to understand the trade-off among accuracy, size, and compute across

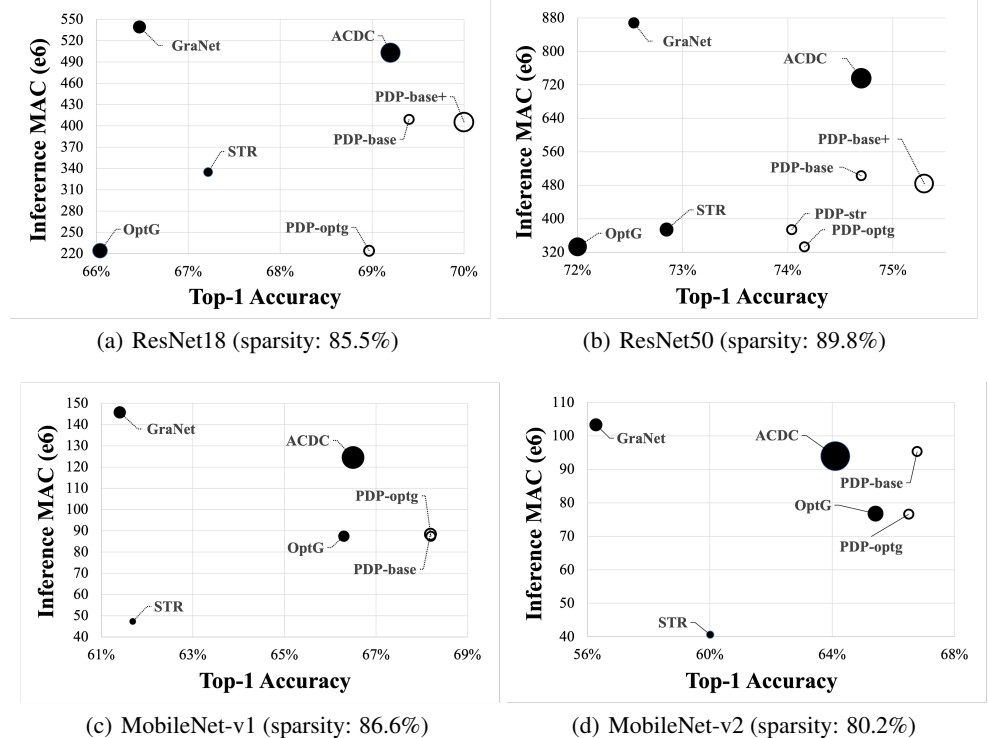

(a) ResNet18 (sparsity: 85.5%)    (b) ResNet50 (sparsity: 89.8%)

(c) MobileNet-v1 (sparsity: 86.6%)   (d) MobileNet-v2 (sparsity: 80.2%)

**Figure 5:** PDP-powered pruning (in white box markers) delivers the Pareto superiority to the other schemes (i.e., the top-bottom corner is the best trade-off) for ResNet18, ResNet50, and MobileNet-v1/v2 on ImageNet1k. The size of markers indicates the relative training overheads. The detailed numbers are in Table 4.

various algorithms, but it still captures the latency benefit on some hardware platforms (see C in Appendix for details). In our experiments with ImageNet1k, all layers, including both the first and last layers, are pruned without any restriction. Also, we estimated the algorithmic overhead and training efficiency by the total monetary cost for GPUs on commercial cloud spot instances [7].

We applied not only the proposed hyper-parameters from the authors but also a set of further fine-tuned hyper-parameters for competing methods (see Table 8 in Appendix for details). Also, since each algorithm used a different number of epochs and showed results at different sparsity levels, **a)** we ran **STR** first to set the target sparsity levels for all the networks for fair comparisons, because all other schemes can control the sparsity level precisely, **b)** we trained ResNet18/50 for 100 epochs and MobileNet-v1/v2 for 180 epochs following [28, 34, 40, 57] except **STR** (which diverged with more epochs for MobileNet-v1/v2). For **PDP**, we fixed the target sparsity for each layer based on the global weight magnitude at the epoch 16 and started pruning at the rate of 1.5% of the target sparsity per epoch for all the experiments which correspond to $s = 16$ and $\epsilon = 0.015$ in Algorithm 1 in Appendix. For detailed experiment configurations, please refer to Table 8. Every experiment began with a randomly initialized model (i.e., no pre-trained model). For **PDP**, we had the following variants to show the value of **PDP** with the same training overhead or per-layer pruning budgets.

- **PDP-base** globally computes the target sparsity by $abs(W)$ at epoch 16 across all the layers.
- **PDP-base+** is the same as **PDP-base** yet with more epochs to match the GPU cost of **ACDC**.
- **PDP-str/optg** uses the per-layer sparsity from **STR/OptG** as input to normalize the MAC.

Our experimental results are highlighted in Fig. 5, where the size of circles indicates the relative training overhead due to pruning. Note that we used only one single node with 8 GPUs due to the limitation in the official implementations for **GraNet** and **OptG**, thus both have the advantage of not having the inter-node communication cost. Also, each approach imposes a different level of training-time overhead, mainly due to the various complexities of training flow and pruning itself as captured in Fig. 5. Overall results can be summarized as follows:

- **PDP** showed the best the model accuracy: **PDP-base** on ResNet18 delivered 69% Top-1 accuracy which is superior to other schemes but at higher MAC than only **STR** and **OptG**.

| Network Sparsity | Method | Top-1 (%) | GPU$ cost($) | MAC (×e6) | Network Sparsity | Method | Top-1 (%) | GPU$ cost ($) | MAC (×e6) |
|---|---|---|---|---|---|---|---|---|---|
| | Dense | 69.8 | 167 | 1814.1 | | Dense | 76.1 | 248 | 4089.2 |
| | GMP | 65.2 | 217 | 263.5 | | GMP | 73.6 | 483 | 419.0 |
| | DNW | 64.4 | 206 | 263.5 | | DNW | 70.7 | 466 | 419.0 |
| | GraNet* | 66.0 | 198 | 539.6 | | GraNet* | 72.5 | 321 | 868.0 |
| ResNet18 | STR | 66.7 | 171 | 334.6 | ResNet50 | STR | 72.8 | 417 | 373.7 |
| 85.5% | OptG* | 65.5 | 277 | 223.7 | 89.8% | OptG* | 72.1 | 591 | 333.0 |
| | ACDC | 68.7 | 356 | 502.8 | | ACDC | 74.7 | 635 | 735.6 |
| | PDP-base | 69.0 | 169 | 408.6 | | PDP-base | 74.7 | 325 | 502.8 |
| | PDP-base+ | **69.5** | 336 | 405.1 | | PDP-base+ | **75.3** | 610 | 483.0 |
| | PDP-str | 68.6 | 169 | 334.7 | | PDP-str | 74.0 | 325 | 373.7 |
| | PDP-optg | 68.5 | 174 | 223.7 | | PDP-optg | 74.2 | 332 | 332.9 |
| | Dense | 70.9 | 277 | 568.7 | | Dense | 71.9 | 297 | 300.8 |
| | GraNet* | 61.4 | 367 | 145.7 | | GraNet* | 56.3 | 439 | 103.4 |
| MobileNet-v1 | STR | 61.7 | 176 | 47.2 | MobileNetv-2 | STR | 60.0 | 285 | 40.6 |
| 86.6% | OptG* | 66.3 | 340 | 87.4 | 80.2% | OptG* | 65.4 | 545 | 76.8 |
| | ACDC | 66.5 | 641 | 124.5 | | ACDC | 64.1 | 812 | 93.9 |
| | PDP-base | **68.2** | 281 | 88.3 | | PDP-base | **66.8** | 354 | 95.3 |
| | PDP-str | 65.3 | 307 | 47.2 | | PDP-str | 60.7 | 307 | 40.6 |
| | PDP-optg | 68.2 | 297 | 87.3 | | PDP-optg | 66.5 | 343 | 76.6 |

$ the GPU cost ($) is based on a commercial cloud spot instance pricing.
* used only one with 8 GPUs due to the limitations in the public code.
◇ when PDP applied for MobileNet-v3 (5.5M parameters) with an 80% sparsity target, we achieved 71.5% top-1 ImageNet1K accuracy, which is 2.5% down from the dense version.

**Table 4:** PDP compared with other unstructured pruning algorithms on ImageNet1K shows the best trade-off among accuracy, inference MAC, and training overheads. More results are available in Section F in Appendix.

- **PDP** offered the better model accuracy for a given pruning target: With the custom sparsification target for each layer, **PDP-str/optg** demonstrated the 2-3% higher Top-1 accuracy at the same MAC, demonstrating the effectiveness of the proposed method.

- When we use the similar GPU budget for additional epochs with **ACDC** which is noted as **PDP-base+**, our method further improved the Top-1 accuracy from 69% to 69.5% for ResNet18 and from 74.7% to 75.3% for ResNet50 with slight fewer MACs.

**Random Pruning for NLP Benchmark:** We compared **PDP** with the state-of-the-art pruning results from **MVP** [44] and **POFA** [56] (quoted from the respective papers) in addition to **OptG** and **STR** (reproduced in our environment) on a BERT model [10] for the two largest NLP tasks of the GLUE benchmark, MNLI (Multi-Genre Natural Language Inference) with 392,702 samples and QQP (Question-answering Natural Language Inference) with 363,836 samples. Following the setups in **MVP** and **POFA**, we used the same batch size 32 per GPU (i.e., global mini-batch size of 512), excluded the embedding and the last linear layer from pruning, and trained the *bert-base-uncased* from HuggingFace from scratch with self-distillation. The teacher for the distillation is from the best checkpoint in the first 40 epochs. We use the weights of 0.95 on the distillation loss and 0.05 on the task loss for **PDP**, and 0.75 on the distillation loss and 0.25 on the task loss for **STR**. We could not use distillation for **OptG**, as it caused the out-of-memory error due to the extra-parameter overheads from both pruning and distillation. Our experimental results in Table 5 can be summarized as follows:

- **STR** underperforms on all the best cases even though it could not achieve the target sparsity.

- **OptG** shows better model accuracy only than **GMP** and worse than others in our setup.

- **PDP** outperformed all other methods for both MNLI and QQP.

**Structured/Channel Pruning for Vision Benchmarks:** We compared PDP-driven N:M pruning and channel pruning on the ImageNet1k dataset [9]. For N:M pruning, we reproduced the **LNM** [59] results using the public code base but without the color augmentation to keep the experimental environment normalized. For **PDP**, we simply reused the hyper-parameters and configurations as in Table 8 in our Appendix, and the top-1 accuracies by various N:M configs with ImageNet1k on ResNet18/50 are presented in Table 6. We can observe that **PDP** outperforms **LNM** on all the test

| Benchmark | Metric | | Methods | | | | | | |
|---|---|---|---|---|---|---|---|---|---|
| | | | Dense | STR* | OptG | GMP | MVP | POFA | PDP |
| MNLI matched | accuracy (%) | 90 | 84.5 | 75.8 | 78.5 | n/a | 81.2 | 81.5 | **83.1** |
| | | 94 | | 74.4 | 76.9 | 74.8 | 80.7 | n/a | **82.0** |
| MNLI mismatched | | 90 | 84.9 | 76.3 | 78.3 | n/a | 81.8 | 82.4 | **83.0** |
| | | 94 | | 74.1 | 76.5 | 75.6 | 81.2 | n/a | **82.4** |
| QQP | accuracy (%) | 90 | 91.2 | n/a | 89.8 | n/a | 90.2 | 90.9 | **91.0** |
| | f1 | 90 | 88.1 | n/a | 86.2 | n/a | 86.8 | 87.7 | **88.0** |

\* Since **STR** cannot control the sparsity precisely, we report the metrics with our closest achieved sparsity levels, 85.9% for the 90% case and 91.8% for the 94% case.

**Table 5: PDP** delivers the best accuracies on the BERT-base model with the MNLI (Multi-Genre Natural Language Inference) and QQP (Quora Question Pairs2) tasks of the GLUE benchmark.

| Network | Method | Batch size | #epochs | N:M | | | | avg GPU cost ($) |
|---|---|---|---|---|---|---|---|---|
| | | | | 2:4 | 4:8 | 1:4 | 2:8 | |
| ResNet18 | LNM | 256 | 120 | 69.6 | **70.2** | 65.1 | 68.4 | 395 |
| | PDP | 1024 | 100 | **70.2** | 70.1 | **68.7** | **69.1** | 275 |
| ResNet50 | LNM | 256 | 120 | 74.6 | 75.1 | 74.1 | 75.0 | 812 |
| | PDP | 1024 | 100 | **75.9** | **75.8** | **75.0** | **75.3** | 380 |

\* For **LNM**, we used the the public dense model from TorchVision for channel pruning, as the original baseline from [59] (which has higher accuracy) is not available publicly. For the example of ResNet50 on ImageNet1k, **LNM** used a dense model with 77.3% top-1 accuracy while the public dense model in TorchVision) yields 76.1% top-1 accuracy.

**Table 6:** Structured Pruning: **PDP** can be directly to do N:M pruning due to its generality. PDP delivers the superior results than the latest N:M pruning in [59] at 46% less training cost.

| Method | Batch size | #epochs | Top-1 (%) | MAC drop (%) |
|---|---|---|---|---|
| NISP | ? | 90 | 75.3 | 44.0 |
| DCP | 256 | 60 | 75.0 | 55.0 |
| SCP | 256 | 100 | 75.3 | 54.3 |
| PDP | 1024 | 100 | **75.9** | 54.9 |

\* **SCP, DCP**, and **NISP** reported only ResNet50 results with MAC drop instead of sparsity. Hence, for **PDP**, we report the nearest MAC drop we obtained (54.9%) at 57% channel sparsity.

**Table 7:** Channel Pruning: the generality of **PDP** helps deliver the state-of-the-art results without modifications.

cases, even with 4x larger batch size in 20 fewer epochs. **LNM** training cost is also much higher than **PDP** because of its costly weight regularization and complex back-propagation scheme.

For channel pruning, we compared **PDP** with **SCP** [26], **NISP** [55], and **DCP** [62]. Note that **SCP** uses the $\beta$ in BatchNorm to select the channels to prune (i.e., $beta \leq \epsilon$), hence applicable to limited types of networks only. We again reused the hyper-parameters and configurations as in Table 8 in our Appendix for **PDP**, and the top-1 accuracy with ImageNet1k on ResNet50 is reported in Table 7. We can see that **PDP** can be used for channel pruning and show superior performance out of the box.

# 5 Conclusion

We showed that a simple and universal pruning method PDP can yield the state-of-the-art random/structured/channel pruning quality on popular computer vision and natural language models. Our method requires no additional learning parameters, yet keeps the training flow simple and straightforward, making it a practical method for real-world scenarios. We plan to extend our differentiable pruning into quantization, making both jointly differentiable and optimizable by the task-loss.

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

| Network | Method | Batch size | #epochs | #wu | #GPUs | #Nodes | Main optimizer, scheduler | Mask optimizer, scheduler | other params |
|---|---|---|---|---|---|---|---|---|---|
| ResNet20 | CS | 128 | 85x5 | 0 | 1 | 1 | SGD 0.9 0.1, multi-step 56,75 by 0.1 | same as the main | init value: -1 |
| | PDP | 128 | 120 | 5 | 1 | 1 | SGD 0.9 1e-4, cosine 0.4 | n/a | $\tau : 1e^{-4}, \epsilon : 0.015$ |
| GPT2 | OptG | | | | | | Refer to [1] | | |
| | PDP | | | | | | Refer to [1] | | |
| ResNet18 | GraNet | 128 | 100 | 5 | 8 | 1 | SGD 0.9 1e-4 (0.0 for BN), cosine 0.1 | n/a | init density: 0.5 |
| | GMP | 128 | 100 | 5 | 16 | 2 | SGD 0.875 1e-5, cosine 0.256 | same as the main | |
| | STR | 256 | 100 | 5 | 16 | 2 | SGD 0.875 2.251757813e-5, cosine 0.256 | same as the main | init value: -3200 |
| | OptG | 256 | 100 | 5 | 16 | 2 | SGD 0.9 1e-4, cosine 0.1 | SGD 0.9 0, cosine 0.1 | $\beta : 1.0$ |
| | ACDC | 256 | 100 | 5 | 8 | 1 | SGD 0.875 3.05175781 3e-4, cosine 0.256 | n/a | 8 alterations |
| | PDP-base/str/optg | 1024 | 100 | 5 | 16 | 2 | SGD 0.9 1.4e-5, cosine 1.8 | n/a | $\tau : 1e^{-4}, \epsilon : 0.015$ |
| | PDP-base+ | 1024 | 200 | 5 | 16 | 2 | SGD 0.9 1.4e-5, cosine 1.8 | n/a | $\tau : 1e^{-4}, \epsilon : 0.015$ |
| ResNet50 | GraNet | 128 | 100 | 5 | 8 | 1 | SGD 0.9 1e-4 (0.0 for BN), cosine 0.1 | n/a | init density: 0.5 |
| | GMP | 128 | 100 | 5 | 16 | 2 | SGD 0.875 1e-5, cosine 0.256 | same as the main | |
| | STR | 256 | 100 | 5 | 16 | 2 | SGD 0.875 2.251757813e-5, cosine 0.256 | same as the main | init value: -3200 |
| | OptG | 256 | 100 | 5 | 16 | 2 | SGD 0.9 1e-4, cosine 0.1 | SGD 0.9 0, cosine 0.1 | $\beta : 1.0$ |
| | ACDC | 256 | 100 | 5 | 8 | 1 | SGD 0.875 3.05175781 3e-4, cosine 0.256 | n/a | 8 alterations |
| | PDP-base/str/optg | 1024 | 100 | 5 | 16 | 2 | SGD 0.9 1.4e-5, cosine 1.8 | n/a | $\tau : 1e^{-4}, \epsilon : 0.015$ |
| | PDP-base+ | 1024 | 200 | 5 | 16 | 2 | SGD 0.9 1.4e-5, cosine 1.8 | n/a | $\tau : 1e^{-4}, \epsilon : 0.015$ |
| MobileNet-v1 | GraNet | 128 | 180 | 5 | 8 | 1 | SGD 0.9 1e-4 (0.0 for BN), cosine 0.1 | n/a | init density: 0.5 |
| | STR | 256 | 100 | 5 | 16 | 2 | SGD 0.875 3.751757813e-5, cosine 0.256 | same as the main | init value: -12800 |
| | OptG | 256 | 180 | 5 | 16 | 2 | SGD 0.9 4e-4, cosine 0.1 | SGD 0.9 0, cosine 0.1 | $\beta : 1.0$ |
| | ACDC | 256 | 180 | 5 | 8 | 1 | SGD 0.875 3.05175781 3e-4, cosine 0.256 | n/a | 8 alterations |
| | PDP-base/str/optg | 1024 | 180 | 5 | 16 | 2 | SGD 0.9 1.4e-5, cosine 1.8 | n/a | $\tau : 1e^{-4}, \epsilon : 0.015$ |
| MobileNet-v2 | GraNet | 128 | 180 | 5 | 8 | 1 | SGD 0.9 1e-4 (0.0 for BN), cosine 0.1 | n/a | init density: 0.5 |
| | STR | 256 | 100 | 5 | 16 | 2 | SGD 0.875 3.751757813e-5, cosine 0.256 | same as the main | init value: -12800 |
| | OptG | 256 | 180 | 5 | 16 | 2 | SGD 0.9 4-e4, cosine 0.05 | SGD 0.9 0, cosine 0.05 | $\beta : 1.0$ |
| | ACDC | 256 | 180 | 5 | 8 | 1 | SGD 0.875 3.05175781 3e-4, cosine 0.256 | n/a | 8 alterations |
| | PDP-base/str/optg | 1024 | 180 | 5 | 16 | 2 | SGD 0.9 8e-6, cosine 0.8 | n/a | $\tau : 1e^{-4}, \epsilon : 0.015$ |
| Bert | GMP | | | | | | Refer to Section 6 and Table 3 in [44] | | |
| | MOV | | | | | | Refer to Section 6 and Table 3 in [44] | | |
| | POFA | | | | | | Refer to Section E and Tables 5 and 6 in [56] | | |
| | STR (85.9%) | 32 | 140 | 5 | 16 | 2 | AdamW 1e-8, multiplicative 1e-4 0.95 | SGD 0.875 0.06, cosine 1e-4 | init value: -90 |
| | STR (91.7%) | 32 | 140 | 5 | 16 | 2 | AdamW 1e-8, multiplicative 1e-4 0.95 | SGD 0.875 0.065, cosine 1e-4 | init value: -110 |
| | OptG | 32 | 140 | 5 | 16 | 2 | AdamW 1e-8, cosine 1e-4 | SGD 0.9 0, cosine 1e-4 | $\beta : 1.0$ |
| | PDP | 32 | 140 | 5 | 16 | 2 | AdamW 1e-8, cosine 1e-4 | n/a | $\tau : 1e^{-4}, \epsilon : 0.015$ |

**Table 8:** The hyper-parameters in Sections 3 and 4.

SGD: momentum, weight decay, cosine: learning_rate, AdamW: epsilon, multiplicative: learning_rate, gamma, #wu: the number of warm-up epochs.

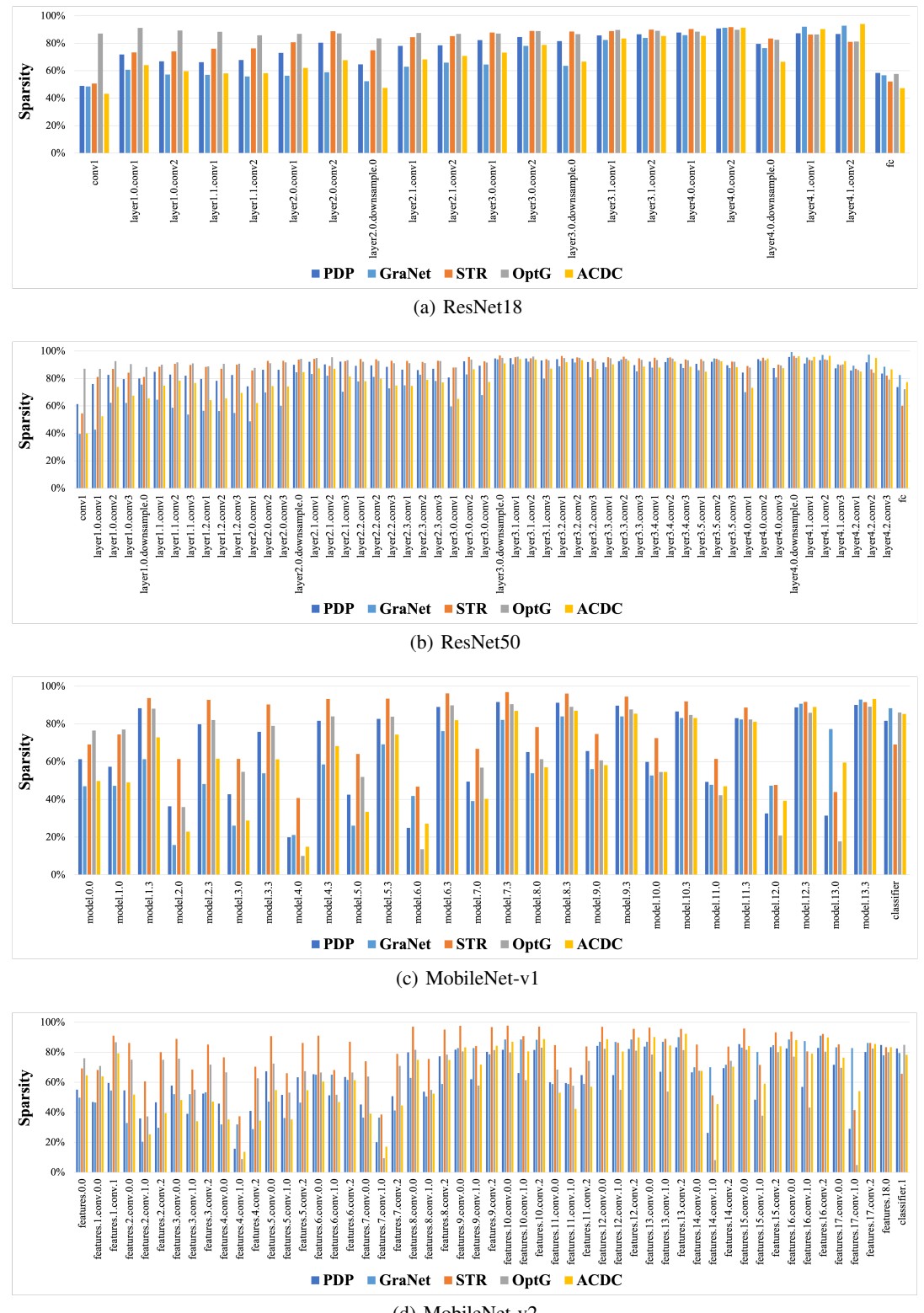

**Figure 6:** Layer-wise sparsity allocation from the experiments in Table 5.

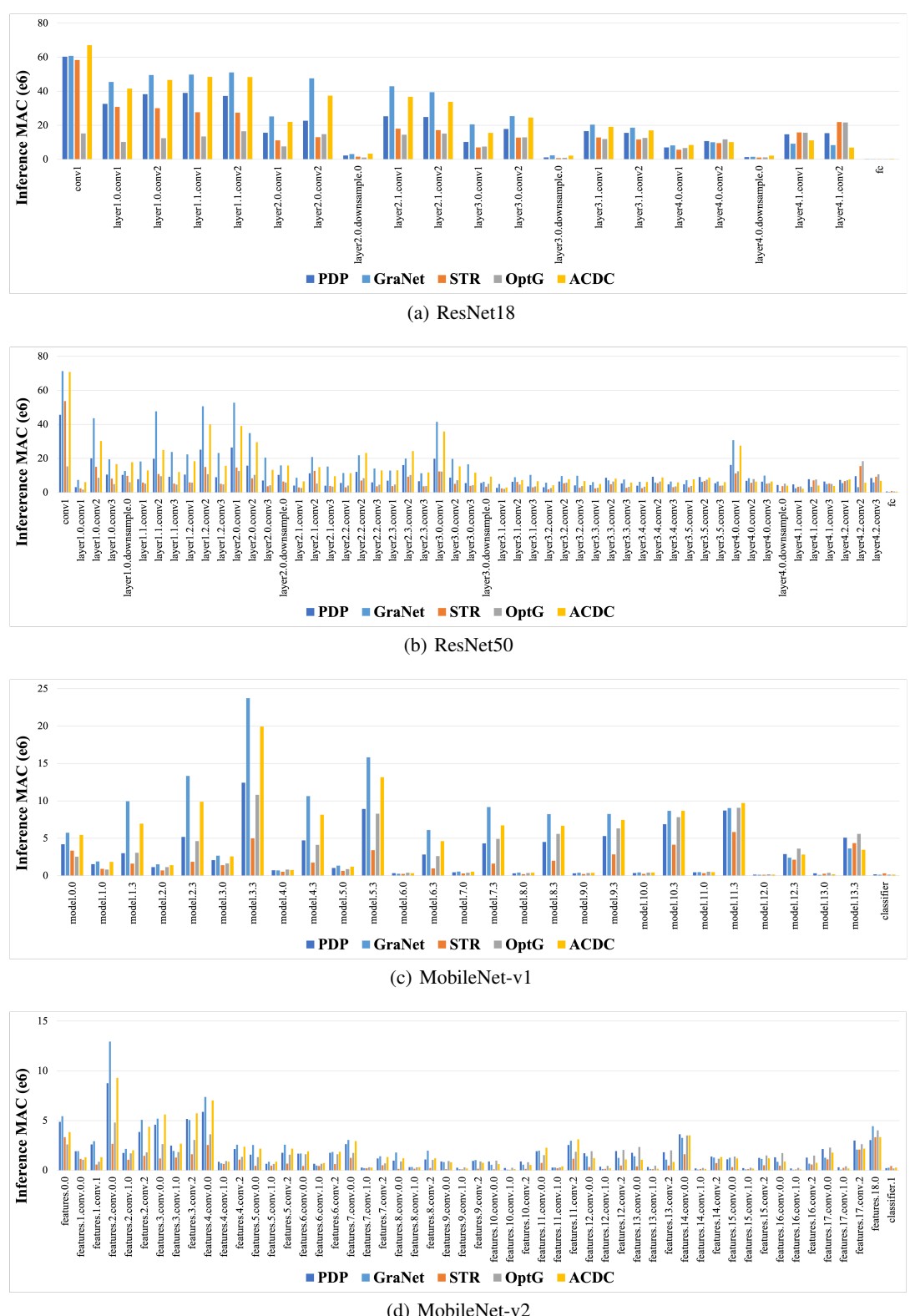

**Figure 7:** Layer-wise Inference MAC distribution from the experiments in Table 5.

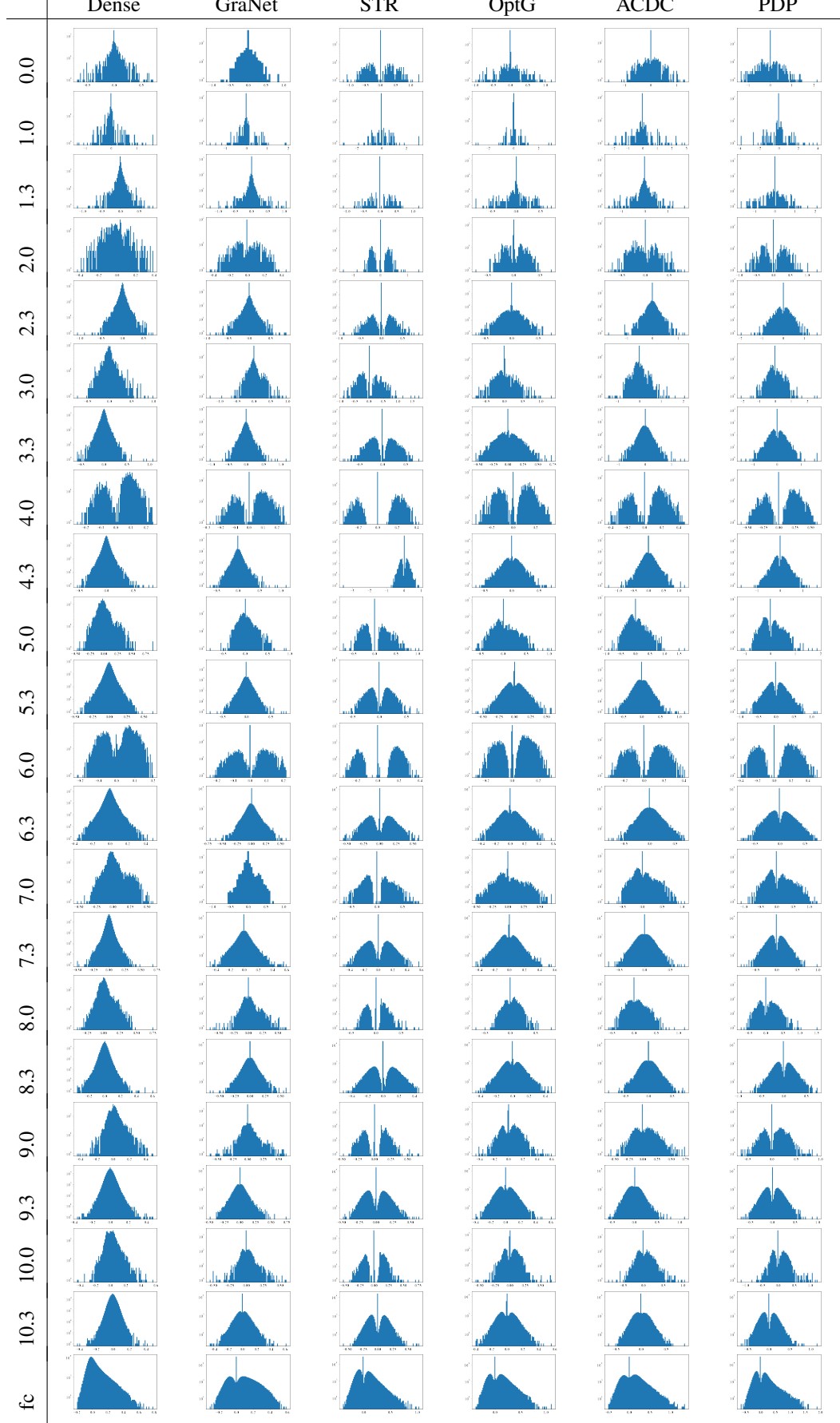

**Table 9:** The weight histograms in log scale for MobileNet-v1 in Table 5.

# A   Training Configurations and Hyper-parameters

Since some techniques in Sections 3 and 4 require extra training parameters and pruning scheduling as shown in Table 1, we disclose the training configurations and hyper-parameters we found the best in Table 8.

# B   Trade-off in Pruning

Pruning for DNN requires exploring a good trade-off between model accuracy and inference latency under a given pruning target. Such a challenge can be elaborated with the MobiletNet-v1 Dense case in Fig. 8 where the following observations can be made:

- The earlier layers have significantly fewer parameters than the later layers while still having comparable inference MACs as shown in (a). For example, the final classifier, which is a linear layer, has the lowest inference MAC but the 2nd largest parameters.
- When per-parameter inference MAC is computed as in (b) (which is in log-scale), we can easily see that the parameters in the earlier layers get much more involved in the inference than those in the later layers. For example, the MAC-per-parameter for the last classifier is only 1.

Then, with a given pruning target, one pruning scheme can favor heavily pruning the classifier, as it is easier to hit the target without affecting model accuracy much (i.e., each parameter shows up only once in the forward pass), but this would fail to reduce the inference MAC enough. Then, the other scheme may favor aggressively pruning the earlier layers to significantly minimize the inference latency at a much greater risk of degrading the model accuracy. Therefore, it is critical to find a good balance between accuracy and inference speed. According to our experimental results, PDP can accomplish such a balance using differential pruning w.r.t. the task loss. Such trade-off can be optimized differently depending on whether a particular sparsity pattern or structure is enforced.

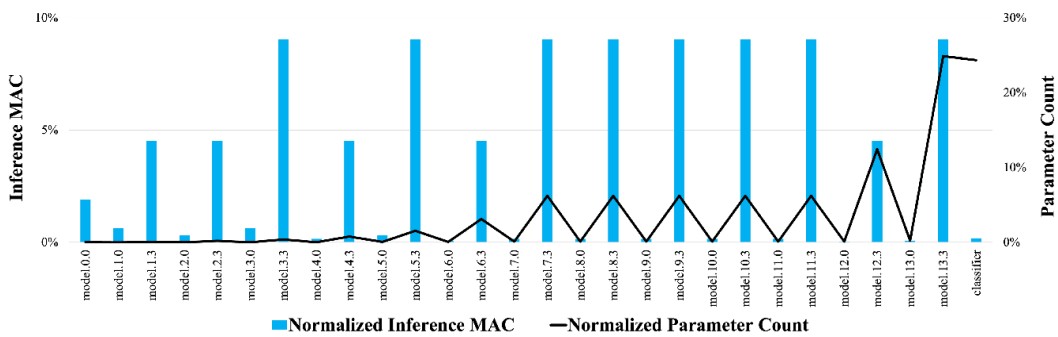

(a) Normalized inference MAC and parameter count for each layer.

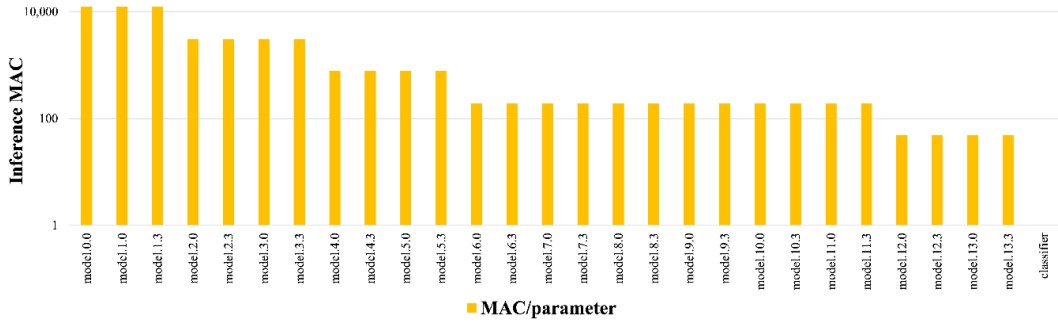

(b) The inference MAC per parameter for each layer.

**Figure 8:** Layer-wise Inference MAC and Parameters from the MobileNet-v1 Dense case in Table 5.

**Random Pruning**: Unstructured schemes make individual and independent pruning decision for each weight to maximize the flexibility and minimize the accuracy degradation. Simple and gradual/iterative pruning based on the weight magnitude has been studied extensively [13, 15, 19, 61]. In these schemes, once a weight is pruned, it does not have the *second* chance to become unpruned and improve the model quality. To address such challenges, RigL [12] proposes to grow a sparse network by reallocating the removed weights based on their dense gradients. Applying brain-inspired neurogeneration (i.e., unpruning some weights based on gradients) and leveraging pruning plasticity is proposed [34]. Altering the phase of dense and sparse training to accomplish co-training of sparse and dense models is studied, which results in good model accuracies on vision tasks [40]. Unlike other magnitude-driven pruning, supermask training [60] integrated with gradient-driven sparsity is proposed in [57], where accumulated gradients are used to generate binary masks and straight-through estimator [5] is used for backward propagation. Based on the lottery hypothesis [13], pruning in one-shot with heuristics [47] or gradient-driven metrics [49] is explored. More advanced techniques based on the second-order sensitivity analysis for language models are proposed in [27].

**Structured/Channel Pruning**: Unstructured pruning limits inference latency speedup as it suffers from poor memory access performance, and does not fit well on parallel computation [3, 35]. Recent research extends unstructured pruning by imposing a particular sparsity pattern during pruning at the cost of lower model predictive power, but increases the hardware performance during inference. One popular and effective form of structured pruning is channel pruning, where some channels with negligible effects on the model accuracy are discarded [21, 25, 30]. Using regularization to prune weights in a block is proposed in [29]. [26] leverages the $\beta$ in BatchNorm to select the channels to prune (i.e., $beta \leq \epsilon$) with ReLU assumed, which limits its applicability to wider set of DNNs. N:M pruning enforces that there are N non-zero weights out of every consecutive M weights [59] which enables a compact memory layout and efficient inferences on hardware [24, 37, 59]. A non-differentiable method for N:M pruning with complex back-prorogation based on STE is presented in [59].

## C  Latency reduction with Random Sparsity

While MAC (to measure the latency gain) is rather a theoretical metric, not a measured metric, but it still captures the speedup benefit from pruning to some degree on some modern smartphones where ML accelerators natively support unstructured sparsity [2]. When we ran our pruned model on iPhone12 with the latest OS update, we obtained the following end-to-end latency measurements.

| Method | ResNet50 | MobileNet-v2 |
|---|---|---|
| Dense | 2.71 | 0.95 |
| PDP | 1.39 | 0.75 |
| Speedup (%) | 94 | 27 |

**Table 10:** A single image classification latency in $msec$ on iPhone12 for the pruned models in Table 4.

Due to other system overheads, the end-to-end latency reduction is not as significant as MAC saving, yet our result still implies the potential benefits of unstructured sparsity on a modern smartphone.

## D  PDP Algorithm and Training Flow

In order to obtain $t$ in Fig 2, PDP needs a target pruning ratio $r$. The pruning ratio can be computed by selecting the top weights with larger magnitudes across all the layers and then instantly convert the selections into the per-layer ratios. Another way is to handcraft per-layer ratios, or reuse an existing configuration. Also, PDP is using the softmax operation which makes the *softness* concentrated over the weights around the $t$ (as shown in Fig. 2). Hence, we gradually increase the target pruning ratio from 0 to $r$ so that all low magnitude weights in the pruning range have a chance to use a soft-mask and settle down smoothly. For that purpose, we introduce a scaling step $\epsilon$ to let each weight have opportunities to leverage a soft-mask at least once, which leads to the training flow in Algorithm 1.

We select $r$ to avoid pruning decisions being dominated by the initial weight values (which can be simply random). Choosing the right number of epochs as $r$ depends on various factors such as the

---
**Algorithm 1** Training flow for PDP
---

1: **procedure** TRAIN($\epsilon, s, r, W = [W_0, W_1, ...]$)
2:    **for** epoch $e$ in $[0, 1, 2, s)$ **do**
3:       **for** each mini-batch **do**
4:          forward with $[W_0, W_2, ...]$
5:          backward-pass and update $W$
6:       **end for**
7:    **end for**
8:    $W_p = topK(-abs(W), r \cdot n(W))$
9:    $[r_0, r_1, ...] = [\frac{n(W_p \cap W_0)}{n(W_0)}, \frac{n(W_p \cap W_1)}{n(W_1)}, ...]$
10:   **for** epoch $e$ in $[s, s+1, s+2, ...]$ **do**
11:      $[\hat{r_0}, \hat{r_1}, ...] = min(1, \epsilon \cdot (e - s)) \cdot [r_0, r_1, ...]$
12:      **for** each mini-batch **do**
13:         **for** $i \in \{0, 1, ...\}$ **do**
14:            $W_h = topK(abs(W_i), (1 - \hat{r_i}) \cdot n(W_i))$
15:            $W_l = bottomK(abs(W_i), \hat{r_i} \cdot n(W)_i)$
16:            $t_i = 0.5\{min(W_h) + max(W_l)\}$
17:         **end for**
18:         **for** $i \in \{0, 1, ...\}$ **do**
19:            $[Z_i, M_i] = softmax(\frac{[t_i^2 \mathbb{J}, W_i \circ W_i]}{\tau})$ //element-wise
20:            $\hat{W}_i = M_i \circ W_i$ //element-wise
21:            forward-pass with $\hat{W}_i$
22:         **end for**
23:         backward-pass and update $W$
24:      **end for**
25:   **end for**
26:   $W_i = \lfloor M_i \rfloor \circ W_i, \forall i \in \{0, 1, ..\}$
27: **end procedure**
---

network size, the target sparsity and so on. Hence, $r$ is a user hyper-parameter. In our experiments, the following guidelines worked best (and this is how we selected it).

- $r$ must be larger than the number of warm-up epochs.
- After $r$ epochs, the validation accuracy hits consistently over the half of the accuracy upper-bound (which is 50% for classifications) for 5 epochs.

In lines 2-7, a normal training is performed for the first $s$ epochs. Then, in lines 8 and 9, the per-layer target pruning ratio is computed by selecting the bottom $r \cdot n(W)$ weights globally in terms of the magnitude. Then, in the remaining epochs, we use PDP to generate soft-masks as in the line 15, while gradually increasing the target ratio as in lines 10 and 11. The updated weight distribution is captured by updating $t_i$ as in the line 16 for all weight matrices. Once the entire training is over, we binarize the last mask for each weight to output the fully pruned weight for inference as in the line 26. Overall, the average runtime complexity of PDP is $O(W)$, as we only need to exercise $topK$ algorithm (i.e, sorintg $W$ is not necessary).

While fewer learnable parameters will reduce the model footprint and the communication cost (thus speeding up the training) in a distributed training setup, it does not necessarily lead to smaller memory consumption as the soft mask from PDP needs to generate related activation tensors. To understand the memory consumption better, we made a small GPT2 case with $block\_size = 128$ and $n\_layer = 3$ (a downscale version of the one in Table 2). And, then GPU memory is measured at three different spots for PDP and OptG, and reported in Table 11.

- **Spot0**: right after the model/optimizer are created (the parameter and PyTorch overheads)
- **Spot1**: right after forward, before backward (the peak memory consumption)
- **Spot2**: right after backward, before weight update (other parameter-related overheads)

From Table 11, we can see the following:

| Method | Spot0 | Spot1 | Spot2 |
|--------|-------|-------|-------|
| OptG | 2.93 | 18.3 | 3.58 |
| PDP | 1.30 | 16.3 | 2.93 |
| Saving | 1.63 | 2.0 | 0.65 |

**Table 11:** Memory footprint reduction from PDP.

- Not having extra mask parameters helps to reduce the model/optimizer overheads.

- The peak memory is dominated by the activations (both data and mask). Yet, about 10 saving in the peak memory is observed.

- After backward, the gradients for the learnable masks take up a large memory space, which have been all-reduced in the multi-GPU setting.

To isolate the benefit of the proposed soft mask in PDP from the gradual pruning in Algorithm 1, we ran PDP for ResNet50 and MobileNet-v1 training in the exactly same configuration as in Section 4 without the proposed soft mask, and obtained the following results. The result supports the efficacy of the proposed soft mask in PDP: without it, the ImageNet top1 accuracy drops by 1.6% and 1.4%, respectively.

| Method | ResNet50 | MobileNet-v1 |
|--------|----------|--------------|
| PDP w/o softmask | 73.1 | 66.8 |
| PDP | 74.7 | 68.2 |
| $\Delta$ | 1.6 | 1.4 |

**Table 12:** Top-1 accuracy with ImageNet1k witout the softmax in PDP

# E    Ablation Study: Hyper-Parameter $\tau$ search

In the current PDP implementation, we use a global $\tau$ to control the level of softness in the pruning mask. Therefore, the selection of $\tau$ affects the model predictive power and should be carefully tuned. In order to explore the methodology for the $\tau$ search, we tried various values for MobileNet-v2 training, and the results are plotted in Fig. 9. The selection of $\tau$ affects the model predictive power as shown in Fig. 9 where there appears to be an optimal $\tau$. For examples of MobileNet-v2, $\tau = 1e - 4$ is the best value and is used for all the experiments in Section 4.

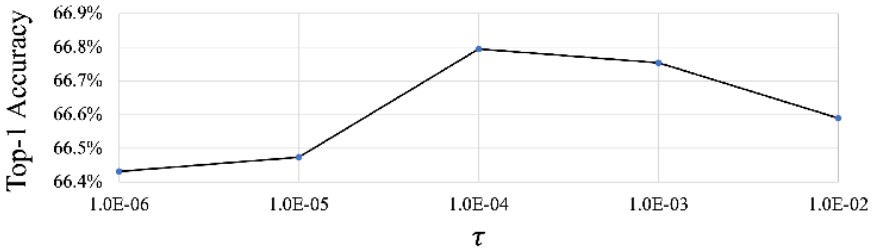

**Figure 9:** MobileNet-v2 with varying $\tau$ values.

Since, Fig. 9 shows a concave curve, one could use a binary search to find the best $\tau$ values w.r.t. the top-1 accuracy. Also, it could be possible to cast $\tau$ as a learnable parameter for each layer or apply some scheduling to improve the model accuracy further (as future work), but still both approaches need an excellent initial point which can be found using a binary search technique.

## F    Additional Result for Section 4.

Different approaches made different sparsity allocations per the characteristics of the algorithm for a given pruning target, which results in complex trade-offs between model accuracy and inference speed. We report the detailed sparsity and inference MAC break-down for each layer in Fig. 6 and Fig. 7 on ImageNet1k and summarize our observations as follows:

- **OptG** prunes the early convolution layers quite aggressively in ResNet18 and ResNet50, which leads to very low inference MACs as shown in Fig. 5 (a) and (b), yet at the cost of the worse Top-1 accuracy. For example, the inference MAC of ResNet18 from **OptG** is more than 2x less than that from **ACDC**,

- Interestingly, **STR** becomes aggressive in pruning the early convolution layers in MobileNet-v1/v2, while **OptG** does not expose such behavior to MobileNet-v1/v2 (unlike it did for ResNet18/50). Such characteristics also favor the low inference latency over the model accuracy. Also, **STR** tends not to prune the last linear layer much as discussed in [28].

- Unlike **OptG** and **STR**, **ACDC** does not prune the early convolution layers much for the tested networks, but prunes somewhat actively for the late convolution and linear layers, which leads to high model accuracies at the cost of higher inference latencies.

- **PDP** is somewhat between **STR** and **ACDC** and modest across all layers in pruning allocation for all the networks, leading to superior accuracy and inference trade-offs. For example, the layers model.13.3 of MobileNet-v1 and features.17.conv.1.0 of MobileNet-v2 have the most difference among algorithms, and **PDP** is modest in pruning these two layers.

- **OptG** has very low inference MACs on the earlier layers of ResNet18 and ResNet50 due to its aggressive pruning on these as seen in Fig. 6 (a) and (b), which leads to the extremely low inference latencies as shown in Fig. 5 (a) and (b).

- **GraNet** tends to prune the earlier layers less but the later layers more in general which explains why **GraNet** shows the highest inference MACs in Fig. 5.

| Network | Method | Sparsity (%) | | | |
|---|---|---|---|---|---|
| | | 80 | 70 | 60 | 50 |
| ResNet-18 | PDP | 69.8 | 70.8 | 71.0 | 71.3 |
| | ACDC | 69.4 | 70.3 | 70.6 | 70.8 |
| MobileNet-v1 | PDP | 69.5 | 71.0 | 71.6 | 71.9 |
| | OptG | 68.1 | 69.1 | 69.6 | 69.7 |
| | ACDC | 68.5 | 69.9 | 70.9 | 71.4 |

**Table 13:** Top-1 accuracy with ImageNet1k: **PDP** outperforms other schemes with various pruning rates.

| Network | Method | Validation dataset | Sparsity (%) | | | |
|---|---|---|---|---|---|---|
| | | | 80 | 70 | 60 | 50 |
| Bert | PDP | matched | 83.7 | 84.0 | 84.3 | 84.7 |
| | | mismatched | 83.4 | 83.8 | 84.4 | 84.5 |
| | OptG | matched | 80.3 | 80.7 | 81.3 | 81.2 |
| | | mismatched | 80.1 | 80.7 | 80.5 | 81.0 |

**Table 14:** Accuracies with MNLI benchmark: **PDP** maintains the similar accuracy lead over other schemes.

Table. 9 shows the pruned weight histograms of MobileNet-v1 from Table 5. We can observe that each algorithm affects the weight distribution in a slightly different way.

- **STR** prefers to split the distribution more widely than others. For the example of the layer 5.0, **STR** clearly separated the positive and negative weights with a wide gap centered at the zero, while others sis not, except **PDP** created a slight dip around the zero to create mild separation.

- **PDP** tends to spread out the sparsified weight distributions more than others. For the example of the fc layer, the weights from **PDP** range from -0.5 to 2.0, while those from

others are from -0.5 to 1.5. On the other hand, **GraNet** tends to keep the weight distributions tight.

We also experimented with varying pruning rates for **PDP, OptG** and **ACDC** for MobiletNet-v1 and ResNet-18 with ImageNet1k, and Bert with MNLI benchmark under the same configurations as in Section 4. Overall, as shown in Table 13 and Table 14, all tested schemes delivered higher accuracy with lower pruning rate, yet we can observe that **PDP** keeps its superiority to other schemes over all the tested pruning rates.

## G   Code References

- **Dense** https://pytorch.org/vision/stable/index.html
- **GradNet** https://github.com/VITA-Group/GraNet
- **OptG** https://github.com/zyxxmu/OptG
- **ACDC** https://github.com/IST-DASLab/ACDC
- **STR** https://github.com/RAIVNLab/STR
- **GMP** https://github.com/RAIVNLab/STR
- **DNW** https://github.com/RAIVNLab/STR
- **CS** https://github.com/lolemacs/continuous-sparsification
- **LNM** https://github.com/NM-sparsity/NM-sparsity

