# OpenReview forum: "PDP: Parameter-free Differentiable Pruning is All You Need"
_NeurIPS.cc/2023/Conference — NeurIPS 2023 poster_

### Official Review · Reviewer_v4x5 · 2023-07-02

**Soundness:** 3 good
**Presentation:** 3 good
**Contribution:** 3 good
**Rating:** 6
**Confidence:** 4

**Summary:**

This paper proposes a parameter-free differentiable pruning, which uses a dynamic function of weights during training to generate soft pruning masks. It can generalize well on random/structured/channel pruning on both vision and NLP tasks. It achieves superior pruning results.

**Strengths:**

1. The proposed PDP approach is novel and interesting, and can generalize well on random/structured/channel pruning.
2. The analysis in Section 3.1 is interesting and provides valuable insights.
3. The proposed PDP approach can be faster than existing differentiable pruning approaches, with smaller pruning accuracy loss.

**Weaknesses:**

1. Instead of learning masks, which would introduce extra trainable parameters, PDP lets the weights of the network adjust themselves and generate soft masks. This approach may seem counterintuitive, as most weights are obtained and converged through expensive training processes. It may be more effective to adjust and learn masks rather than adjusting network weights to generate masks.

2. The paper only provides a simple explanation for why PDP is more effective. A theoretical proof may be more convincing.

3. Although the method is effective on the CNN and BERT models evaluated in this paper, its effectiveness on other models, such as larger vision transformers and LLMs, is still unclear.

4. An important related work[1] is missing.

[1]Xia, Mengzhou, Zexuan Zhong, and Danqi Chen. "Structured pruning learns compact and accurate models." ACL (2022).

**Questions:**

see the weakness section.

**Limitations:**

The main limitation of the paper is that, aside from empirical experimental results, there is no theoretical proof or explanation as to why it is superior to existing non-parameter-free approaches. The solution presented in the paper raises doubts about its effectiveness on larger neural networks, such as ViT and LLMs.

---

> ### Author Rebuttal · Authors · 2023-08-09
>
> **Q0:  This approach may seem counterintuitive, as most weights are obtained and converged through expensive training processes. This approach may seem counterintuitive, as most weights are obtained and converged through expensive training processes. It may be more effective to adjust and learn masks rather than adjusting network weights to generate masks.**
>
> Thank you for the opportunity to reiterate our key contribution. Our paper in nutshell demonstrates that adjusting weights to generate masks is possible and cheaper than the conventional approach (i.e., learn/adjust masks based on the weight values), which is our key contribution. Although it may sound counterintuitive, our comprehensive experimental results support our contributions.
>
> **Q1: The paper only provides a simple explanation for why PDP is more effective. A theoretical proof may be more convincing.**
>
> Thank you for your feedback. Here, we present more explanation on PDP.
>
> For a given gradient on $ \Delta \hat w$, the gradient on $w$ with magnitude-based pruning can be computed as follows:
>
> $ \Delta w_{mag} = \Delta \hat w \cdot m(w) $
>
> In PDP case, the gradient on $w$ is
>
> $ \Delta w_{pdp} = \Delta w_{mag} + 2 \Delta \hat w  \cdot w^2 m(w)(1-m(w))  $
>
> From above, we can observe the followings:
> * $ \Delta w_{pdp} \ge  \Delta w_{mag}$: This implies PDP allows the weight to learn more from each iteration (larger gradient toward loss-decreasing direction), proportionally to its proximity to the $t$.
> * $ \Delta w_{pdp}$ is max when $m(w) \approx 0.5$: This means PDP encourages the weight on the pruning boundary (around $t$) to learn more aggressively, such that it can converge to the better spot faster.
>
> We conjecture these features enable PDP to make a better pruning decision for the weight on the boundary, and provide strong "second" chances to recover from the undesirable weight updates.
>
> **Q2: Although the method is effective on the CNN and BERT models evaluated in this paper, its effectiveness on other models, such as larger vision transformers and LLMs, is still unclear.**
>
> Thank you for the comment. Due to time and resource constraints, we could not experiment with a larger LLM. Yet, the GPT2 results in Table 3 indicate that PDP can offer a better trade-off between LLM training cost and LLM accuracy. Hope this could address some of your concerns.
>
> **Q3: An important related work[1] is missing.**
>
> Thanks for the new reference. We will cite the paper in the updated draft.

---

> > ### Comment · Reviewer_v4x5 · 2023-08-21
> > **Response to authors**
> >
> > Thanks for your further explanation, which answers my questions.  I will increase the score.

---

### Official Review · Reviewer_8xNe · 2023-07-03

**Soundness:** 3 good
**Presentation:** 3 good
**Contribution:** 2 fair
**Rating:** 7
**Confidence:** 4

**Summary:**

This paper deals with the pruning algorithm PDP on DNN, the main innovation is to generate a differentiable mask based on weights using a designed threshold t, and the softmax function. This mask can ensure the gradient propagation during forward and backward process while training, reducing the accuracy loss.

**Strengths:**

1. This paper is well written and easy to follow.
2. The proposed method is novel and provides an efficient way to learn pruning masks without incurring heavy parameter burden.
3. The performance increase remains constant on various sparse ratios and models.

**Weaknesses:**

1. The employment of different masks can impact training process performance, yet a higher SAD[1] may result in a significant decline in accuracy. Given that the masks are still changeable in the later stages of PDP training, how can it provide high accuracy?

[1] Learning N: M fine-grained structured sparse neural networks from scratch. In ICLR, 2020.

2. The description of PDP algorithm in section 3.2 may not be completely unambiguous. For the calculation of the conditional point t, the meaning of the paper should be to take the median of $r*n(W)$ and $(1-r)*n(W)$, but figure 3 (a) shows that $t=0.5\{min(W_h)+max(W_l)\}$, which seems to be wrong.

3. In Table 6, the performance of LNM seems to be quite different from that reported in the original paper. Please explain this.

**Questions:**

Please see the weakness part.

**Limitations:**

No limitations discussed.

---

> ### Author Rebuttal · Authors · 2023-08-09
>
> **Q0: The employment of different masks can impact training process performance, yet a higher SAD[1] may result in a significant decline in accuracy. Given that the masks are still changeable in the later stages of PDP training, how can it provide high accuracy?**
>
> Thank you for the question. Sparse Architecture Divergence (SAD) from [1] defines the changes in the pruning mask during training. The insight behind in SAD is that lower SAD will incur less disruption to the training process, potentially improving the model quality. We first like to point out that our PDP pruning mask is soft and continuous during training, and only get binarized for validation/test/inference. According, in fact, our PDP can make SAD between iterations smaller than the hard/binarized pruning cases.
>
> As an illustration, for the hard masks, the following scenario leads to total SAD 2 and max SAD 1.
>
> |     | iter0  | iter1 | iter2  | iter3 |
> | --- | --- |  --- | --- | --- |
> | Mask | 0 |  1  | 1 |  0  |
> | SAD   |  - |  1 | 0 |  1  |
>
> For PDP case with $t$=0.5, the same scenario could be the following, where total SAD is 0.4 and max SAD 0.2.
>
> |     | iter0  | iter1 | iter2  | iter3 |
> | --- | --- |  --- | --- | --- |
> | training-mask | 0.4 |  0.6  | 0.6 |  0.4  |
> | SAD   |  - |  0.2 | 0 |  0.2  |
>
> which implies PDP slowly tries the same pruning scenario as the above but very gradually using the soft mask.
>
> Hence, PDP will offer much better model stability during training in terms of SAD
> thanks to our proposed differentiable soft mask, and let each weight converge into the right pruning choice at the end. When the network is finalized with a binary mask at the end of the training, our PDP can yield a higher-quality model as shown in Table 6 (i.e. comparison against LNM[1] (which proposed SAD).
>
>
> **Q1:  The description of PDP algorithm in section 3.2 may not be completely unambiguous. For the calculation of the conditional point t, the meaning of the paper should be to take the median of  and , but figure 3 (a) shows that , which seems to be wrong.**
>
> Appreciate the opportunity to clarify the definition of the conditional point $t$. $t$ is essentially a weight value that would have the exact 0.5 mask value. Assuming every weight value in a layer is unique, $W_l \cup W_h$ includes all the weights in the layer. Therefore in this case, $t$ is the median of max($W_l$) and min($W_h$), and also is essentially the mean of the two numbers  as in Fig. 3 (b). Note that $r \cdot n(W)$ and $(1-r) \cdot n(W)$ are to find out $W_l$ and $W_h$ using TopK, so practically the same as $|W_h|$ and $|W_l|$ (i.e., the number of weights).
>
> The example on the right side in Fig. 4 (a) also illustrates the process of finding $t=0.08$ by taking the median (or mid-point) of 0.09 and 0.07.
>
>
>
> **Q2: In Table 6, the performance of LNM seems to be quite different from that reported in the original paper. Please explain this.**
>
> Thank you for the opportunity to clarify the LNM results.  We believe it could be due to two reasons:
> * It is because of different dense baseline models as starting points. For example of ResNet50, LNM used a dense model with 77.3% top-1 accuracy but this starting checkpoint is not publicly available. Hence, we used the public dense model (in torchvision) with 76.1% top-1 accuracy. Such difference in the dense model accuracy contributes to the differences in the accuracies from LNM.
>
> * LNM is using a custom color augmentation according to the public repo. Hence, we removed this feature to keep the data augmentation normalized.

---

### Official Review · Reviewer_tv9j · 2023-07-04

**Soundness:** 3 good
**Presentation:** 1 poor
**Contribution:** 3 good
**Rating:** 5
**Confidence:** 4

**Summary:**

The paper focuses on DNN pruning and proposes a novel approach using a soft mask during training. The soft mask is designed to encourage weights around the pruning threshold to actively switch their states, aiding in the recovery of pruned weights. This method is simple, efficient, and introduces no additional parameters. Experimental evaluations of the proposed approach have been conducted on ResNets, MobileNets, and BERT. It also explores different pruning settings such as N:M pruning, random pruning and channel pruning.

**Strengths:**

* The paper proposes a soft mask during training, which smooths the participation of weights around the pruning threshold in the neural network. This can give more chances for weights around pruning threshold to change their states, especially help pruned weights actively recover themselves.
* The compared methods are enough in experiments.
* The paper discusses relevant works and provides a comprehensive analysis.

**Weaknesses:**

* Method:
  *  Lack of differentiable analysis: The title and content of the paper stress the differentiable pruning, however, method part  (3.2) does not give differentiable analysis.
  * Insufficient explanation of the design of "t" and calculation of "m(w)": Why is “t” in PDP training flow designed in this way? The paper lacks clear explanations about the design of “t” and calculation of “m(w)”.
  * Lack of analysis and ablation study on the efficacy of the proposed soft mask: I find that the paper adopts progressive pruning as illustrated in the algorithm chart in supplementary material. As progressive pruning is an effective technique for final accuracy, I am doubtful about the real effect of the proposed method. To better understand the effectiveness of the proposed soft mask, we need more analysis and ablation studies.
* Experiments:
  * More experiments about Transformer architectures are preferred. The paper provides BERT models experiments on MNLI dataset. Experiments with ViT models or BERT models on the whole GLUE benchmark will be preferred.
  * In the main body, there are not enough sparsity ratio selections. Table 4 provides only one sparsity ratio for each model. More sparsity ratios, ranging from low to high are necessary to understand the effect of the proposed method.
* Writing:
  * Redundancy between Figure 5 and Table 4: It seems that Figure 5 and Table 4 convey the same information. What is the difference between Figure 5 and Table 4 or why illustrate the same information twice? I think each table or figure should contain specific information, especially in the main body.
  * Improper organization of the paper: The paper first illustrates the benefits of PDP over existing pruning approaches, then introduces the PDP algorithm, which can be confusing. Before readers know the method clearly, it would be difficult to understand its benefits. The better solution would be to change their order or to reconsider the effects of section 3.1, such as serving it as a motivation section.



**Questions:**

Please see the method and experiment points in the weakness part. If my concerns are addressed, I would like to increase my score.

**Limitations:**

There is no potential negative societal impact of their work.

---

> ### Author Rebuttal · Authors · 2023-08-09
>
> **Q0: Insufficient explanation of the design of "t" and calculation of "m(w)": Why is “t” in PDP training flow designed in this way? The paper lacks clear explanations about the design of “t” and calculation of “m(w)”.**
>
> Thank you for the chance to clarify the important notation in our work.
>
> $t$ is the weight value which will have exactly 0.5 chance to be pruned and 0.5 chance to be not-pruned of a layer. And, we capture $t$ from the weight distribution of a layer as in Fig. 3 (a). Consequently, $t$ serves as a conditional point to decide whether a weight will have a soft mask $m(w)$ larger than 0.5.
>
> The way we compute $m(w)$ is differentiable using softmax, as shown in Fig. 3 (a). More specifically, we compute $m(w)$ as follows:
>
> $m(w) = \frac{e^{\frac{w^2}{\tau}}}{e^{\frac{w^2}{\tau}}+e^{\frac{t^2}{\tau}}} $
>
> Thus, when $w$ happens to be the same as $t$, $m(w)$ is 0.5. If $w$ is relatively larger than $t$, then $m(w)$ is also larger than 0.5 as in Fig 3. (b).
>
> **Q1:The title and content of the paper stress the differentiable pruning, however, method part (3.2) does not give differentiable analysis.**
>
> Thank you for the opportunity to explain more details. Without loss of generality, let's assume $\tau = 1$ (the temperature of softmax). Then, the masked weight $\hat w$ in PDP is
>
> $\hat w = m(w) \cdot w = \frac{e^{w^2}}{e^{w^2}+e^{t^2}} \cdot w $
>
> Then, for a given gradient on $\hat w,  \frac{\partial L}{\partial \hat w}$,
>
> $\frac{\partial L}{\partial w} = \frac{\partial L}{\partial \hat w} m(w)  + 2\frac{\partial L}{\partial \hat w} w^2 m(w)(1-m(w)) $
>
> The 1st term is a typical gradient in mask-based pruning, and the 2nd term is an additional gradient with a positive factor. Then,
> * if $m(w)  \approx 0$ or hard-prune, $\frac{\partial L}{\partial w} = 0$ likewise other pruning algorithms.
> * if $m(w)  \approx 1$ or hard-not-to-prune,  $\frac{\partial L}{\partial w} = \frac{\partial L}{\partial \hat w}$ likewise other pruning algorithms.
> * when $m(w)  \approx 0.5$ (i.e., pruning decision is unclear), $m(w)(1-m(w))$ is maximized, boosting the $w$ movement.
>
> Hence, PDP will accelerate the SGD updates for the weights near the pruning boundary ($t$) toward a loss-decreasing direction (which means more learning from each iteration), encouraging the weights to settle with the proper pruning decision at the end. Even if the current gradient is not globally beneficial for the task, many "second" chances will eventually help recover the damages in an accelerated manner. We will add this differentiable analysis to the final draft.
>
>
> **Q2: Lack of analysis and ablation study on the efficacy of the proposed soft mask**
>
> Thank you for the feedback. Gradual pruning is strong and popular; in fact, GMP, STR, ACDC, GradNet, and OptG all use some form of gradual pruning. To analyze the benefit of the proposed soft mask in PDP as ablation study, we re-run PDP for resnet50 and mobilenet_v1 training in the exactly same configuration as in Section 4 **without**  the proposed soft mask.
> |     | ResNet50  | MobileNet_v1 |
> | --- | --- | --- |
> | PDP w/o softmask   | 73.1 |  66.8 |
> | PDP | 74.7 |  68.2 |
> | $\Delta$| 1.6  | 1.4 |
>
> The result supports the efficacy of the proposed soft mask in PDP: without it, the ImageNet top1 accuracy drops by 1.6% and 1.4%, respectively.
>
> **Q3: More experiments about Transformer architectures are preferred. The paper provides BERT models experiments on MNLI dataset. Experiments with ViT models or BERT models on the whole GLUE benchmark will be preferred.**
>
> Thank you for suggesting helpful experiments. While we couldn't complete the whole GLUE benchmark due to the limited time and resources, we were able to perform additional experiments with the QQP (Quora Question Pairs2) benchmark which is
> * The 2nd largest dataset in GLUE (363,846 samples), following MNLI (392,702)
> * MVP reported only MNLI and QQP (two largest) out of the GLUE benchmarks.
> * MVP and POFA reported the results on QQP, thus making comparison possible.
>
> We used the same hyper-params as in Section 4 and targeted 90% sparsity, and obtained the following results (including MNLI results from the current submission).
>
> |     | OptG |MVP  | POFA | PDP|
> | --- | --- | -- | --- | --|
> | MNLI-m    | 78.5 |81.2 |  81.5 | 83.1|
> | MNLI-mm   | 78.3 |81.8 |  82.4 | 83.0|
> | QQP-acc   | 89.8 | 90.2 |  90.9 | 90.9 |
> | QQP-f1   | 86.2 |86.8 |  87.7 | 87.7 |
>
> The result shows that PDP can offer the state-of-the-art 90.9% accuracy for QQP which is already very close to a dense model accuracy of ~91%. POFA showed the same quality results but using a more complex training flow.
>
> **Q4: In the main body, there are not enough sparsity ratio selections. Table 4 provides only one sparsity ratio for each model. More sparsity ratios, ranging from low to high are necessary to understand the effect of the proposed method.**
>
> Thank for the suggestion. In Tables 10/11 of Appendix, we have MobileNet-v1, ResNet18, and BERT results on the sparsity targets ranging from 50% to 80%. We will mention these additional results clearly in Section 4.
>
> **Q5: Redundancy between Figure 5 and Table 4**
>
> Thank you for the review. Indeed, some redundant information exists between Fig. 5 and Table 4, because they are complementary in the sense that one for high-level abstraction and the other for low-level details. The numbers in the Table couldn't abstract out the complex trade-off among accuracy-latency-cost in an intuitive manner. Hence, we added Fig. 5 with key numbers to visualize the trade-off and help readers on this complex matter.
>
> We agree that the space in the main body can be better utilized for new information/results (especially ones from this rebuttal), and we accommodate your feedback in the final draft.
>
> **Q6:  The better solution would be to change their order or to reconsider the effects of section 3.1**
>
> Thank you for the feedback. We will reorder as suggested and polish the write-up in the updated draft.

---

> > ### Comment · Reviewer_tv9j · 2023-08-18
> > **Post-Rebuttal**
> >
> > Thanks for the detailed response.
> >
> > I appreciate the author's explanation about Q1 and experiments. Thus I would increase the score to borderline accept. But I suggest the author to improve the writing and give more explanation like the response of Q1 in the paper.

---

### Official Review · Reviewer_R2q9 · 2023-07-06

**Soundness:** 3 good
**Presentation:** 3 good
**Contribution:** 3 good
**Rating:** 7
**Confidence:** 4

**Summary:**

This paper describes a novel pruning algorithm named parameter-free differentiable pruning (PDP). The core idea is to generate soft pruning masks (i.e., mask values are not finalized until the final iteration of fine-tuning) using a parameter-free, differentiable dynamic function of the weights of the network. This approach eliminates the need to train additional mask parameters (and related hyper-parameter tuning) and appears to be easy to integrate into existing training pipelines. PDP is evaluated on networks drawn from vision and NLP, and on both unstructured and structured N:M sparsity patterns. The paper reports both accuracy and MAC improvements and provides a comparison of PDP to other SOTA pruning algorithms.


**Strengths:**

* The paper is fairly well-written, but could use some reorganization to help with the overall flow.
* The paper targets the relevant and important problem of structured sparsification of large DNNs to improve inference efficiency.
* While soft and progressive pruning are well-known in the literature, its parameter-free nature and the specific mask computation make PDP unique.
* Evaluation methodology is sound, with relevant comparisons to other SOTA approaches, and results being reported on various types of networks (CNNs, attention-based, etc.) across disparate domains.

**Weaknesses:**

Measuring inference efficiency in MACs can often be misleading. For instance, unstructured pruning can reduce MACs by getting rid of individual ineffectual computations, but this is difficult to realize in modern parallel hardware such as GPUs and TPUs. Reporting at least some data points with actual runtime numbers would be useful.


**Questions:**

* What is the motivation for selecting the per-layer sparsity value (r) after a few epochs of training? How would a user select the right number of epochs for a given network,  sparsity pattern and global sparsity degree?
* Readability suggestion: consider moving Section 3.2 before 3.1.


**Limitations:**

Limitations have been adequately addressed in the paper.

---

> ### Author Rebuttal · Authors · 2023-08-09
>
> **Q0: Measuring inference efficiency in MACs can often be misleading. For instance, unstructured pruning can reduce MACs by getting rid of individual ineffectual computations, but this is difficult to realize in modern parallel hardware such as GPUs and TPUs. Reporting at least some data points with actual runtime numbers would be useful.**
>
> Thank you for your feedback on MAC. We agree that MAC is rather a theoretical metric, not a measured metric, but it still captures the speedup benefit from pruning to some degree. GPUs don't support unstructured sparsity, but there exist modern smartphones where ML accelerators natively support unstructured sparsity (unfortunately, the link with details cannot be provided per the NeurIPS rebuttal policy). We used a 2-year old phone with the latest OS update and obtained the following latency measurements in msec.
> |     | ResNet50  | MobileNet_v2 |
> | --- | --- | --- |
> | Dense | 2.71 |  0.95  |
> | PDP   |  1.39 |  0.75 |
>
> Due to other system overheads, the end-to-end latency reduction is not as significant as MAC saving, yet still shows the potential benefits of unstructured sparsity on a modern device. We hope these data points would address the reviewer's concern .
>
> **Q1: What is the motivation for selecting the per-layer sparsity value (r) after a few epochs of training? How would a user select the right number of epochs for a given network, sparsity pattern and global sparsity degree?**
>
> Thank you for the question. $r$ is selected after a few epochs to avoid pruning decisions being dominated by the initial weight values (which can be simply random). Choosing the right number of epochs for a given network depends on various factors the reviewer has already mentioned. Hence, this is a part of hyper-parameter tuning. In our experiments, the following guidelines worked best (and this is how we selected it).
> * passes the warm-up epochs
> * hits consistently over the half of the accuracy upper-bound (which is 50% for classifications) for 5 epochs.
>
> **Q2: Readability suggestion: consider moving Section 3.2 before 3.1.**
>
> Thank you for the suggestion. We will accommodate in the updated draft.

---

> > ### Comment · Reviewer_R2q9 · 2023-08-21
> >
> > Thank you for responding to my questions. My score remains the same.

---

### Official Review · Reviewer_qvwL · 2023-07-10

**Soundness:** 2 fair
**Presentation:** 3 good
**Contribution:** 2 fair
**Rating:** 5
**Confidence:** 2

**Summary:**

The paper introduces a new DNN pruning scheme called Parameter-free Differentiable Pruning (PDP), which is an efficient and effective train-time pruning method that offers state-of-the-art qualities in model size, accuracy, and training cost. Unlike existing pruning approaches, PDP generates soft pruning masks for weights in a parameter-free manner, making it easy to apply to various vision and natural language tasks, DNN architectures, and structured pruning constraints. The paper presents experimental results on several benchmark datasets, including MobileNet-v1 and BERT, demonstrating that PDP achieves impressive results in terms of model size, accuracy, and training cost.

**Strengths:**

- The paper introduced a PDP, a novel differentiable pruning methods that is parameter-free, which uses a dynamic function of weights to generate soft pruning masks for the weights.
- PDP can be applied tto structured and channel pruning, such as N:M pruning, where top-of-the-line GPUs support such configuration.

**Weaknesses:**

- The PDP differentiable pruning does not introduce extra parameter, but it still need to generate (soft) mask from weight, which would induce extra activation maps, how is the memory consumption during differentiable pruning compare to other SoTAs?
- The paper seem provide results on MobileNet-v1/v2 but not on MobileNet-v3, can the author elaborate on why?

**Questions:**

As shown in weaknesses part.

**Limitations:**

The paper did not any (Peak) memory consumption result of PDP during differentiable pruning.

---

> ### Author Rebuttal · Authors · 2023-08-09
>
> **Q0: The PDP differentiable pruning does not introduce extra parameter, but it still need to generate (soft) mask from weight, which would induce extra activation maps, how is the memory consumption during differentiable pruning compare to other SoTAs?**
>
> Thank you for raising an important question regarding memory consumption. While fewer learnable parameters will reduce the model footprint and the communication cost (thus speeding up the training) in a multi-node setup, we agree that still extra activation maps will be induced. To understand the memory consumption better, we made a small GPT2 case with block_size=128 and n_layer=3 (a down-scale version of the one in Table 3). And, then GPU memory (in GB) is measured at three different spots for PDP and OptG (i.e., other SoTAs).
> * **Spot0**: right after model/optimizer are created (the parameter + pytorch overheads)
> * **Spot1**: right after forward, before backward (the peak memory consumption)
> * **Spot2**: right after backward, before weight update (other parameter-related overheads)
>
> |     | Spot0  | Spot1 | Spot2 |
> | --- | --- | --- | --- |
> | OptG | 2.93 |  18.3  | 3.58 |
> | PDP   | 1.30 |  16.3 | 2.93 |
>
> From the table, we can see the following:
> * Not having extra mask parameters helps to reduce the model/optimizer overheads.
> * However, the peak memory is dominated by the activations (both data and mask). And as pointed out by the reviewer, even a soft mask still requires activation space. Yet, about 10% saving in the peak memory is observed.
> * After backward, the gradients for the learnable masks take a large memory space, which have been all-reduced in the multi-GPU setting.
>
> We will add the above discussion in the final draft.
>
> **Q1: The paper seem provide results on MobileNet-v1/v2 but not on MobileNet-v3, can the author elaborate on why?**
>
> We appreciate your feedback on MobileNet-v3. We didn't experiment with MobileNet-v3, because the prior arts don't report the related results, thus making comparison with other SoTAs on MobileNet-v3 hard.
>
> When we applied PDP with an 80% sparsity target for MobileNet-v3 (5.5M parameters), we achieved 71.5% top-1 ImageNet1K accuracy, which is 2.5\% down from the dense version. We will include this result in the final draft.

---

### Author Rebuttal · Authors · 2023-08-10

We like to thank the reviewers and ACs for the help and feedback. The highlight of our rebuttal includes the following.

New experimental results:
- Peak memory consumption is measured with PDP and OptG on a small GPT2 model.
- PDP result with MobileNet-v3 and ImageNet1k is added.
- Latency benefit with unstructured sparsity is measured on a modern smartphone.
- The efficacy of the proposed soft mask is studied with ResNet50 and MobileNet-v1 on ImageNet1k.
- PDP and other techniques have been experimented with the QQR dataset of GLUE benchmark.

New analysis:
- A theoretical aspect of PDP is presented.
- PDP is explained in the context of SAD.

---

### Decision · Program_Chairs · 2023-09-21

**Decision:**

Accept (poster)

**Comment:**

Although the reviewers highlighted several strengths of the paper, such as the interest of the problem tackled, the novelty of the method, the good discussion of the literature, and the soundness of the experiments, they also initially raised some concerns regarding the theoretical analysis of the method, some aspects of the experiments, and the clarity of some parts of the paper. The authors' feedback was well received by the reviewers, who eventually reached a consensus for acceptance. The AC nonetheless recommends the authors to update their manuscript based on their feedback and double-check the writing as mentioned by Reviewer tv9j.